# Bacterial Biofilm Formation Using PCL/Curcumin Electrospun Fibers and Its Potential Use for Biotechnological Applications

**DOI:** 10.3390/ma13235556

**Published:** 2020-12-06

**Authors:** Daniella Alejandra Pompa-Monroy, Paulina Guadalupe Figueroa-Marchant, Syed G. Dastager, Meghana Namdeo Thorat, Ana Leticia Iglesias, Valentín Miranda-Soto, Graciela Lizeth Pérez-González, Luis Jesús Villarreal-Gómez

**Affiliations:** 1Facultad de Ciencias de la Ingeniería y Tecnología, Universidad Autónoma de Baja California, Blvd Universitario 1000, Unidad Valle de Las Palmas, Tijuana 22260, Baja California, Mexico; daniella.pompa@uabc.edu.mx (D.A.P.-M.); figueroa.paulina@uabc.edu.mx (P.G.F.-M.); aiglesias@uabc.edu.mx (A.L.I.); perez.graciela@uabc.edu.mx (G.L.P.-G.); 2Facultad de Ciencias Químicas e Ingeniería, Universidad Autónoma de Baja California, Universidad #14418, Parque Internacional Industrial Tijuana, Tijuana 22390, Baja California, Mexico; 3Academy of Scientific and Innovative Research (AcSIR), AcSIR Headquarters CSIR-HRDC Campus, Postal Staff College Area, Sector 19, Kamla Nehru Nagar, Ghaziabad 201002, India; sg.dastager@ncl.res.in; 4NCIM Resource Center, CSIR-National Chemical Laboratory, Pune 411008, India; mn.thorat@ncl.res.in; 5Tecnológico Nacional de México/Instituto Tecnológico de Tijuana/Centro de Graduados e Investigación en Química, Calzada Del Tecnológico S/N, Fraccionamiento Tomas Aquino, Tijuana 22414, Baja California, Mexico; vmiranda@tectijuana.mx

**Keywords:** bacteria, curcumin, electrospinning, *Escherichia coli*, *Staphylococcus aureus*, *Pseudomona aeruginosa*

## Abstract

Electrospun nanofibers are used for many applications due to their large surface area, mechanical properties, and bioactivity. Bacterial biofilms are the cause of numerous problems in biomedical devices and in the food industry. On the other hand, these bacterial biofilms can produce interesting metabolites. Hence, the objective of this study is to evaluate the efficiency of poly (Ɛ- caprolactone)/Curcumin (PCL/CUR) nanofibers to promote bacterial biofilm formation. These scaffolds were characterized by scanning electron microscopy (SEM), which showed homogeneous fibers with diameters between 441–557 nm; thermogravimetric analysis and differential scanning calorimetry (TGA and DSC) demonstrated high temperature resilience with degradation temperatures over >350 °C; FTIR and ^1^H-NMR serve as evidence of CUR incorporation in the PCL fibers. PCL/CUR scaffolds successfully promoted the formation of *Escherichia coli*, *Staphylococcus aureus* and *Pseudomonas aeruginosa* biofilms. These results will be valuable in the study of controlled harvesting of pathogenic biofilms as well as in metabolites production for biotechnological purposes.

## 1. Introduction

Bacterial biofilms are a multicellular life phase in which the cells are sessile and live in biofilms; they represent an important medical problem due to the toxicity of some of them contaminating the surfaces of medical devices and promoting nosocomial bacterial infection in patients [1]. Most importantly, one of the most predominant characteristics that contribute to the severity of biofilm infections is their peculiar resistance to antibiotics [2,3,4]. Biofilms are defined as surface-associated microbial heterogeneous structures, comprising different populations of microorganisms surrounded by a self-produced matrix, which allows their attachment to inert or organic surfaces [5]. Nonetheless, the mechanisms of the onset of biofilm formation are poorly understood [6]. In contrast, biofilm formation has gained much interest in in the bioremediation industry [7], i.e., *Pseudomonas sp.*, biofilm, among others, has demonstrated enhanced crude oil degradation [8] and wastewater bioremediation [9].

Curcumin (CUR), known as (bis-1,7-[4-hydroxy-3-methoxyphenyl]-hepta-1,6-dione), is a hydrophobic polyphenolic pigment which is found in the rhizomes derived from *Curcuma longa* and the yellow pigment known as turmeric [10]. CUR has many medical applications due to its biological properties such as anti-inflammatory, antioxidant, antitumor, antifungal and antibacterial, and burn wound healing effects [10,11,12,13]. Despite all of its applications, CUR has the disadvantage of poor water solubility and low permeability [14]. Subsequently, for scaffold applications, CUR traditionally has been electrospun with the help of a polymeric nanocarrier, mainly for controlled drug release, as can be observed in Table 1. 

Electrospun nanofibers have a myriad of applications, such as tissue engineering [16,17], for example in skin regeneration [18], wound dressing [19], cartilage regeneration [20], and bone regeneration [21], among others. Applications also include sensors [22,23], drug delivery systems [24,25], filters [26,27], and solar cells [28,29], among others.

It is important to search for some specific characteristics of the fibers that can be useful for improving the growth of industrially important bacteria in order to produce, in a faster way, interesting metabolites. Some desirable characteristics are a tridimensional structure, large surface area, nutrient disposition availability, and controlled degradation of fibers [30]. In previous work, we reported electrospun nanofibers that can be used for the growth of pharmaceutical drugs producing bacteria, such as actinomycetes [31]. Moreover, few reports that proposed the use of nanofibers as bacterial biofilms appeared in literature; i.e., electrospun cellulose acetate nanofiber membranes for *Lactobacillus plantarum* (*L. plantarum*) present a biofilm formation [32]. In the present study, we will show the capacity of poly (caprolactone)/curcumin (PCL/CUR) to serve as tridimensional scaffolds, that improved the biofilm formation of *Staphylococcus aureus, Escherichia coli*, and *Pseudomonas aeruginosa* for biotechnological purposes.

## 2. Materials and Methods

### 2.1. Materials

*Curcuma longa L.* leaf was donated by Dr. Syed Dastager from the National Collection of Industrial Microorganism (NCIM), CSIR-National Chemical Laboratory, Pune-41008, Maharashtra, India. Poly (caprolactone) (PCL) (24,090 g/mol of molecular weight MW), ethanol and chloroform (ACS from Fermont) were purchased from Aldrich (St. Louis, MO, USA), and used as received.

### 2.2. Solution Preparation

Separate solutions were prepared in order to incorporate CUR into a PCL solution. First, a polymeric solution of 10% PCL was prepared using chloroform as a solvent, following the methodology of Ruckh et al. [33]. Solution was stirred with 100 rpm at 70 °C until the polymer dissolved. CUR ethanolic extract was prepared by cutting and grinding the CUR leaves until a powder was obtained, then; 10 mL of ethanol was added to 10 mg of weighed CUR powder and the solution was stirred with 60 rpm at 35 °C for 30 min until a homogeneous solution was evident. Ethanolic CUR solutions were added at different proportions (2.0, 2.5, 5.0, and 10.0%) to PCL solutions at room temperature (~24 °C), with constant stirring at 100 rpm for 2 h according to Bui et al. [34], in order to find the highest CUR concentration that can be added to PCL solution which leads to fiber formation (Table 2).

### 2.3. Preparation of PCL/CUR Electrospun Fibers

The standard electrospinning technique was used to prepare PCL/CUR to nanofibers. Polymer solutions were loaded into a syringe that was set over a pump injection system, then the electrospinning device (UABC, Tijuana, México) was set up with the following conditions: temperature: 20–22 °C, % humidity: 32–43%, distance from syringe tip to collector: 10 cm, voltage: 20 kV and flow rate: 1 mL/h. 

### 2.4. Preparation of PCL/CUR Cast Films 

PCL and PCL/CUR casting films were prepared using PCL and PCL/CUR solutions obtained in Section 2.2. The membranes were obtained by casting 1.5 mL of each solution in a circular mold (diameter 2.5 cm), followed by the evaporation of solvent at room temperature according to Brianezi et al. [35]. 

### 2.5. Characterization of PCL/CUR Microfibers

#### 2.5.1. Scanning Electron Microscopy (SEM)

The morphology and diameter of the prepared fibers were determined by scanning electron microscopy (SEM); a small section of fibrous material was placed in an SEM. A field emission microscope JEOL JSM 7600F (JEOL Ltd., Tokyo, Japan) with an accelerating voltage of 20 kV was used for images. Images were taken at 400×, 3000× and 12,000× of magnification. Software Image J was used to measure the fiber diameter and the percentage of pore area in the membranes.

#### 2.5.2. Fourier Transform Infrared (FTIR)

Fourier transform infrared spectroscopy by attenuated total reflectance (ATR) was done on a Thermo Scientific Nicolet 6700 (Thermo Fisher Scientific, Waltham, MA, USA). The spectra were collected at 20 °C in the range of 4000–500 cm^−1^. Four sweeps were performed at a resolution of 4 cm^−1^.

#### 2.5.3. Hydrogen Nuclear Magnetic Resonance (^1^H NMR)

^1^H NMR spectra were acquired at 400 MHz with a Bruker Avance III spectrometer (Bruker, Billerica, MA, USA) at 30 °C. Chemical shifts were reported in ppm and referenced to residual solvent resonance.

#### 2.5.4. Thermogravimetric Analysis (TGA)

TGA was used to measure the change in mass by increasing the temperature, thermal stability, and high degradation temperature of all samples, performed under thermogravimetric analysis. The test was carried out using a heating period of 10 °C/min from room temperature (±20 °C) to 500 °C in a flask platinum unsealed using nitrogen with a flow of 20 mL/min by using Shimadzu model TGA-Q500 (Shimadzu Scientific Instruments Incorporated, Kyoto, Japan). The mass of the samples analyzed varied between 5 to 10 mg. This technique allowed us to determine the temperature at which thermal degradation begins (T_onset_) and the change in mass per temperature increases. Thermogravimetric curves (DR_TG_) were conducted to identify the maximum degradation temperature (T_deg max_).

#### 2.5.5. Differential Scanning Calorimetry (DSC)

Samples were characterized by DSC using a Shimadzu model DSC-Q100 equipment (Shimadzu Scientific Instruments Incorporated, Kyoto, Japan). Ramps were programmed to heat from room temperature (±20 °C) to 200 °C over a period of 10 °C/min (first run). Subsequently, an unsealed aluminum container was used for each sample with a carrier gas with a flow of 20 mL/min. The mass of the samples analyzed varied from 5 to 10 mg. After the run, the furnace was cooled with liquid nitrogen to a temperature of –20 °C or –30 °C.

### 2.6. Bacterial Growth Assay 

Nanofiber mats and cast film samples were cut with a 0.5 cm diameter using a drill, sterilized on both sides with UV-light radiation for 15 min, and placed at the bottom of a 96 plate-wells. Bacterial strains such as *Staphylococcus aureus* (ATCC 23235), *Escherichia coli* (ATCC 25922), and *Pseudomonas aeruginosa* (ATCC 15442) were previously cultured in a sterile Mueller-Hinton medium for 24 h at 35 °C. Afterward, bacterial strains were standardized to absorbance at 0.5 McFarland, compared with a standard tube (0.132 abs at 620 nm, 1.5 × 10^8^ CFU/mL) with saline solution. Also, 150 μL of clean medium were added to each well in which the fibers and films were placed. Then, 50 μL of each inoculum were added (*E. coli*, *S. aureus*, *P. aeruginosa* and medium). As a negative control, clean media and clean media with fibers and films were used. As a positive control, 150 μL of medium and 50 μL of each strain were placed without fibers or films. Also, the four concentrations of pure CUR were used as controls. All exposed bacterial cells with fibers were incubated for 24 h at 35 °C. As fibers do not dissolve, with the help of sterile forceps, samples were removed and placed in a new well and washed with 200 μL of clean medium, fibers were discarded, and solutions were measured in a Microplate reader (Thermo Fisher Scientific, Waltham, MA, USA) at 620 nm.

### 2.7. Biofilm Formation Study

Samples were cut, prepared, and sterilized as before *(vide supra*) and exposed to the three selected bacterial strains (*Staphylococcus aureus*, *Escherichia coli*, *Pseudomonas aeruginosa).* For the experiment, bacteria were incubated for 12 h at 35 °C in Mueller Hilton media [36], and after incubation, samples were taken out and dried with a vacuum pump and a small section of fibrous material was prepared for SEM images at 400×, 3000× and 12,000×.

### 2.8. Statistical Analysis

The experiments were done in triplicate in an independent manner. The results were expressed as mean ± standard deviation of three independent experiments. Data were evaluated by one-way analysis of variance (one-way ANOVA), using Graph Pad Prism version 6.0c software. The results were considered statistically significant when *p* < 0.05.

## 3. Results

Curcumin (*Curcuma longa L*) (CUR) is an herbaceous plant of the *Zingiberaceas* family native to southwestern India. Its main feature is a strong yellow color. It has a waxy texture and is insoluble in water. The most important chemical components of the curcuma leaf are a group of compounds called curcuminoids (Figure 1), which include curcumin, which have exhibited potent anti-inflammatory and antioxidant activities useful for the medical industry [13]. 

The standard electrospinning technique was used to prepare the PCL/CUR nanofibers. For the present study, the optimal electrospinning conditions were the following: distance from syringe tip to collector: 10 cm, voltage: 20 kV and flow rate: 1 mL/h. Similar conditions for electrospun PCL fibers were described before [33] using a 12% *w/v* PCL solution which was loaded into a glass syringe and fed into a 20-gauge blunt-tip needle by a syringe pump at a rate of 2.3–2.6 mL/h. A high-voltage power supply was used to apply voltage in the range of 18–21 kV to the blunt-tip needle that was positioned 10–11.5 cm from the grounded collector plate. Hence, similar electrospinning conditions in both studies were successful for nanofibers production.

For our study, the maximum concentration of CUR was 10% *v/v*. Ravikumar et al. [12] reported the same concentration and solvent systems for their electrospun CAP/CUR nanofibers.

Successful fiber formation was achieved for all CUR concentrations. The fibers were characterized by FTIR, ^1^H NMR, TGA, and DSC. The description is included in the Appendix A.

### 3.1. Scanning Electron Microscopy (SEM)

SEM was used to determine the average fiber diameter, percentage of porosity of the scaffolds, and morphology of the fibers. From a macroscopic view, the nanofibers were smooth and stable, with adequate elasticity and resistance to deformation at touch; the thickness for all the manipulated mats was 0.3 cm (Figure 2A). PCL/CUR3 and PCL/CUR4 (highest concentration of CUR) membranes were yellow-colored to the naked eye.

All samples presented well-defined fibers, with a smooth surface and no bead presence, PCL/CUR2, PCL/CUR3 and PCL/CUR4 fibers presented a similar size and shape as PCLc (control) (Figure 2D–F); in contrast, PCL/CUR1 fibers were three times as thick as the PCLc fibers, and with a shape that resembled flat fibers (Figure 2C) in contrast with the rest of the samples that presented smother fibers [22]. In the case of PCL/CUR4, some electrosprayed artefacts can be seen on the mats; this may be due to the small variations of temperature and % humidity in the ambient environment. Finally, all fibers were randomly disposed of in the collector, and different sizes can be seen in the SEM pictures.

The average diameter of the fibers was uniform, with a low standard deviation, in the range of 441 to 557 nm (Table 3) for PCL/CUR1 and PCL/CUR fibers. For PCLc fibers, the size was between 500 ± 165 nm with a higher frequency between 320–440 nm. As mentioned before, with exception of PCL/CUR1 fibers (1734 ± 525 nm), PCL/CUR fiber diameters were constant. While PCL/CUR2 fibers sizes were above 709 ± 254 nm, PCL/CUR3 fibers were of 441 ± 154 nm, and PCL/CUR4 fibers presented diameters of 557 ± 161 nm (Table 3). 

Although the % of porosity is important in tissue engineering research, particularly in relation to mechanical properties and ideal pore size for cell growth, no such parameters are reported for bacterial proliferation, especially considering that bacterium are much smaller than human cells. Our results show that there is no apparent relationship between % porosity and bacterial growth, although except for PCL/CUR1, increasing the concentration of CUR decreased the % porosity and bacterial growth (Table 3, Figure 3).

As a final note, as is seen in Figure 2, it is expected that the CUR distribution on the fibers is homogeneously blended along the polymeric fibers because direct blending electrospinning with a standard setup was used [37,38].

### 3.2. Fourier Transform Infrared Spectroscopy

The FT-IR spectra of the samples only show the characteristic bands for the PCL carbonyl at 1726 cm^–1^. This is probably due to the low concentration of CUR in the samples, even at the highest concentration of CUR (PCL-CUR4) (Figure 3).

### 3.3. Hydrogen Nuclear Magnetic Resonance (^1^H NMR)

In the ^1^H NMR spectra of the electrospun PCL control fibers (Appendix A), four characteristic signals at 4.06 ppm for –CH_2_OOC–, 2.30 ppm –OCCH_2_–, and the methylene signals –(CH_2_)– units at 1.63 and 1.36 ppm are observed; assignments were made base on previously reported PCL polymers and co-polymer fibers [39,40,41]. In Figure 4, a stack of ^1^H NMR spectra can be observed for all the fiber samples with increasing amounts of curcumin (Figure 4B–E); no significant difference (ANOVA P < 0.05) with the PCL control was observed, and no signals for the curcumin were observed even at the highest concentration of PCL/CUR4 (27 µM CUR) (Figure 4E).

### 3.4. Thermo Gravimetric Analysis (TGA)

TGA was used to evaluate the thermal behavior of the PCLc and PCL/CUR fibers, as mass loss can be determined via temperature increments [41]. As observed in Figure 5, all samples showed high stability (tolerance) at high temperatures. Independent of the CUR concentration, variations in mass loss are not significant (ANOVA P < 0.05) amongst the samples, and no linear relationship with amount CUR can be observed.

In the TGA spectra, the temperatures corresponding to 10, 50, and 100% weight loss were calculated with TA analysis software (TA instruments) (Table 4). Solvent evaporation was responsible for weight loss between 5–10%. The temperature range corresponding to 50% of weight loss in the samples varied between 394–404 °C, with final degradation between 407–496 °C. A PCL weight loss of 88% reportedly occurred between 376 and 480 °C. 

In our study, the decomposition of CUR is not visible; nevertheless, it can be noticed that the samples with a higher concentration of CUR (PCL/CUR) are less temperature resilient, giving evidence of the presence of CUR in the PCL fibers. Another observation is that PCL beads (PCLb) are more resistant to temperature than the electrospun samples fibers (~10 °C higher). All this data shows how the morphology of PCL and the concentration of CUR into the PCL fibers affect temperature resilience. 

### 3.5. Differential Scanning Calorimetry (DSC)

DSC was used to determine the melting point temperature (T_m_) and degradation temperature (T_d_) of PCL/CUR fibers [42]. Analogous to the TG graph, DSC spectra demonstrated similar thermal behavior between all samples (Figure 6, Table 5).

The melting point of CUR is reported to be at 183 °C [43], but this is not appreciated in the DSC graph, probably due to the small concentration of CUR in the PCL fibers (Table 5).

In the case of our samples, the T_d_ of PCL fibers is around 381 °C, and PCL/CUR are between 377–386 °C, corresponding with the reported T_d_ of PCL of the above literature. On the other hand, the complete degradation of PCL/CUR fibers occurs between 409–423 °C, corresponding to the temperature for maximum degradation (T_max_) of PCL (80,000 g/mol MW). 

### 3.6. Bacterial Growth Assay

Bacterial growth studies are important to determine whether PCLc and PCL/CUR fibers better increased bacterial proliferation compared to PCLc and PCL/CUR films and if these fibrous scaffolds alter the bacterial cells’ viability, since these two parameters impact biofilm formation. Thus, in this work, a Gram-positive bacterium (*Staphylococcus aureus*) and Gram-negative bacteria (*Escherichia coli* and *Pseudomonas aeruginosa*) were exposed for 0.5, 12, and 24 h to all fiber and films samples to evaluate its bacterial cytotoxicity and promotion of biofilm formation.

Figure 7 shows how the presence of the nanofibers does not affect the viability of any of the three tested bacterial strains; all optical densities (O.D.)´s behavior is similar with normal growth during the time (ANOVA P < 0.05). In order to better explain the obtained results, the percentage of proliferation rate was calculated, taking into account the normal growth of each bacteria (bacterial suspension without fibers and films), where O.D. is the optical density obtained at 620 nm after the exposure time.
(1)$$Percentage\;of\;proliferation\;\left(\%\right)\;=\;\frac{O.D.\;sample\;\left(film\;or\;fibers\right)\;\times \;100}{O.D.\;normal\;growth\;\left(control\right)}$$

Comparing the efficiency in the increment of the bacterial proliferation between PCL/CUR fiber and PCL/CUR films for *Escherichia coli*, after 30 min of incubation both fibers and films delay the normal growth, ~7% for fibers and ~30% for films, respectively. Then, after the 12 h, around ~30% of improved growth was observed in the presence of fibers, and a huge reduction of growth of about ~73% was elicited from the films. The same behavior was observed after 24 h of incubation.

Hence, for *E. coli*, PCL/CUR nanofibers clearly increment the proliferation rate and PCL/CUR films alter the normal growth. When comparing between each individual formulation of films and fibers, the most efficient fibrous scaffold was PCL/CUR3, which increased the growth about 37% higher than the normal growth in 12 and 24 h, followed by the PCLc fibers, which enhanced growth about a 32% at 24 h. The less interesting fibrous scaffold was PCL/CUR1 with just 5% more growth than the control at 24 h. 

No clear difference can be seen between PCL/CUR fibers with respect to their bioactive effect when the concentration of CUR was varied. This last result is valuable for assessing the appropriate concentration of CUR, depending on the application and expected results.

In the case of *Pseudomona aeruginosa* growth, a different bacterial behavior was observed compared to the latter results. In general, PCL/CUR films did not affect bacterial growth at any time interval tested (ANOVA P < 0.05). On the contrary, PCL/CUR fibers delay the growth by ~52% after the first 12 h, but after 24 h, all PCL/CUR fibers increase the proliferation rate ~15%; this could be due to an adaptation period. At 24 h, PCL/CUR3 presented the best results with an increment of (~31%), followed by the PCLc fibers with 22% enhanced growth with respect to the control. As with the PCL/CUR films, PCL/CUR2 and PCL/CUR4 do not significatively affect the normal growth of P. aeruginosa.

Finally, PCL/CUR fibers efficiently promoted the growth of *Staphylococcus aureus* compared to PCL/CUR films. At 0.5 h, PCL/CUR fibers promoted ~16% the proliferation rate, and PCL/CUR films reduced the growth by ~11%. Then, at 12 h, PCL/CUR fibers enhanced by about ~80% the cellular replication, whilst PCL/CUR films enhanced just ~5%. Lastly, after 24 h, PCL/CUR fibers promoted the growth by ~36% and PCL/CUR films reduced the normal growth by ~14%. The PCL/CUR fiber with the best results was PCL/CUR1, with ~94% enhanced growth at 12 h, followed by the PCL/CUR3 fibers, which promoted the replication of *S. aureus* by ~86% at 12 h.

It can thus be demonstrated that PCL/CUR fibers are efficient systems that promoted the growth of all the tested bacteria; particularly, these fibrous scaffolds were more effective in promoting growth in *Staphylococcus aureus* and *Escherichia coli*. Bacteria Exposed to PCL/CUR films presented different behaviors depending on the bacterial strain. From all of the PCL/CUR fibers, the system with best results was PCL/CUR3, but no statistically significant difference between the CUR concentrations was found (ANOVA P < 0.05). The behavior of exposed bacteria toward the PCL/CUR scaffolds is dependent on the bacterial strain. Nonetheless, the system with the best results was PCL/CUR3, although no statistically significant difference between the CUR concentrations was found (ANOVA P < 0.05). 

In the present study, to ensure the sterilization process (15 min UV irradiation for both sides), all PCL/CUR fibers and films were exposed to clean media to observe the presence of any growth, but no changes in the O.D. were observed. Moreover, to demonstrate that different CUR concentrations have an effect on the bacteria, as well as confirm the capacity of CUR to cause cellular stress, pure CUR ethanolic extract (dried) at different concentrations (CUR1: 5 µM; CUR2: 7 µM; CUR3: 14 µM and CUR4: 27 µM) and pure CUR leaf were tested in the presence of the three bacterial strains. It can be observed that the unprocessed CUR leaf does not significantly affect the normal growth of each bacteria (ANOVA P < 0.05), but dried CUR ethanolic extracts decreased the growth proportional to the concentration; therefore, the higher the concentration of CUR employed, the higher the reduction of proliferation is in all three bacteria at all times. However, this decrement is not significant different (ANOVA P < 0.05) between CUR concentrations, and a tendency is clearly observed (Appendix A).

Despite our results, CUR has been reported to have bactericidal activity, especially with Gram-positive (*Staphylococcus aureus* and *Enterococcus faecalis*) and Gram-negative (*Escherichia coli* and *Pseudomonas aeruginosa*) bacteria, but the tested CUR was type CUR I (Figure 7). In our study, CUR was loaded into polymeric fibers at very low concentrations (5–27 µM), which explains the low bactericidal activity of the PCL/CUR fibers. Moreover, it was reported that the mode of antimicrobial action of curcumin depends on the delivery system [44]. In our study, no significant difference (ANOVA P < 0.05) in the PCL/CUR activity between Gram-positive (*S. aureus*) and Gram-negative bacteria (*E. coli* and *P. aeruginosa*) was observed; nevertheless, it was reported that Gram-positive bacteria show a significantly higher sensitivity to curcumin than the Gram-negative ones. 

### 3.7. Biofilm Formation Study

This assay was performed as a qualitative test to observe the efficiency of the fibers as bacterial scaffolds for biofilm formation, and thus, it is complementary to the above bacterial growth assay. For that reason, and taking into account the above results, no statistically significant difference in the cellular growth was found, and not many incidences between the concentrations of CUR were definitive in the results. PCL/CUR4 fibers were chosen to show bacterial biofilm formation along the fibers due to their highest quantity of CUR and best appreciation of the bacterial cells posed on the fibers. However, the SEM images of PCL, PCL/CUR1, PCL/CUR2, and PCL/CUR3 are presented as (Appendix A).

Figure 8A–F shows how *E. coli* and *P. aeruginosa* are forming the bacterial biofilm along the PCL/CUR4 fibers during the 12 h of incubation. As other studies have shown, Gram-negative *P. aeruginosa* and Gram-positive *S. epidermidis* bacteria have displayed uninhibited suspended growth in medium exposed to PCL scaffolds after 6 h at 37 °C; moreover, both bacterial species multiplied prolifically and populated scaffold surfaces within dense colonies [33].

*S. aureus* created a very consolidate biofilm (Figure 8G), but it seems to degrade the polymeric fibers after 12 h (Figure 8I). *S. aureus* has been identified as extracellular polymeric substances (EPS) producing bacteria which are capable of degrading several polymeric molecules, with the aim of surface detachment [45]; this observation is an insight of the cellular stress mechanism of *S. aureus*.

Despite this fact, several research groups have studied the biofilm’s formation for 24 h [46,47], and several studies have employed shorter incubation times [48,49]. In the present study, the objective was to observe how bacterial colonies attach to the surface of the fibers and proliferate inside the tridimensional structure of the scaffolds creating the biofilms. However, for future work, we intend to study the formation of bacterial biofilms after 24 h and evaluate its effectivity in several biotechnological application studies.

## 4. Discussion

### 4.1. Scanning Electron Microscopy (SEM)

The diameters of the fibers were similar to those reported by Ruckh et al., [33], where the PCL fiber diameter was between 557 ± 399 nm. The latter PCL fibers were proposed for bone tissue engineering and a drug delivery system of rifampicin. Several studies reported diverse average diameters of fibers for biomedical applications. Yang et al. [50] obtained PCL and PCL-gentamicin fibers with an average of 100–120 nm diameter. In another study, diameters between 1403 ± 660 nm were obtained [51]. Finally, Ramírez-Cedillo et al. [52] reported the highest fiber diameter between 2990 ± 1020 nm, which was deemed suitable for bone tissue engineering. As can be noticed from the above literature and our results, no specific diameter is reported for PCL fibers, which are intended for biomedical applications. Shababdoust et al. [11] reported that the highest diameter for PU/CUR fibers was above 284 ± 112 nm. In the case of CAP/CUR fibers, the size was between 300 ± 20 nm [12]. The average fiber diameter of the PHBV/CUR nanofibers varied from 207 ± 56 to 519 ± 15 nm, depending on the curcumin concentration [13]; almond gum/PVA/CUR nanofibers varied from 121 ± 31 to 169 ± 35 nm, depending also on curcumin concentration [10]. Therefore, as cited above, there is no clear relationship between the concentration of CUR and fiber diameter. 

For our proposed application, it is desirable that fibers degrade or reabsorb into the biofilm media so that in the production and recovery of secondary metabolites no further extraction is needed. The first step of our study is to demonstrate that PCL/CUR scaffolds are useful for bacterial biofilm production. Then, further studies will be necessary to determine the appropriate fiber diameters for the specific application or required metabolite. 

### 4.2. Fourier Transform Infrared Spectroscopy

The PCL fibers’ spectra indicate a strong sharp peak around 1717 cm^−1^, which is due to C=O vibration and at 2942 cm^−1^ which represented a C–H peak [50]. Moreover, several characteristic bands of PCL were found at 2949 cm^–1^ (asymmetric –CH_2_ stretching), 2865 cm^−1^ (symmetric –CH_2_ stretching), 1726 cm^−1^ (carbonyl stretching), 1293 cm^−1^ (C–O and C–C stretching), 1240 cm^−1^ (asymmetric C–O–C stretching), and 1170 cm^−1^ (symmetric C–O–C stretching) [35,36] (Appendix A). 

In the case of CUR, the broad absorption band at 3426 cm^−1^ and the sharp peak at 3508 cm^−1^ are attributed to the phenolic O–H stretching vibration. The bands at 1628 cm^−1^ and 1592 cm^−1^ are related to αβ– unsaturated carbonyl group C=O stretching vibrations. The bands at 1432 cm^−1^ and 1508 cm^−1^ are related to the olefinic bending vibration of the C–H bound to the benzene ring of CUR and C–C vibrations, respectively. The bands at 1277 cm^−1^ and 1154 cm^−1^ are due to C=O stretching and C–O–C stretching modes, respectively [10,12]. Those signals did not appear on the spectra, probably due to the low concentration of CUR. Another limitation of this study is that the degradation of CUR was not evaluated, and a Reversed–Phase High-Performance Liquid Chromatography (RP–HPLC) test is needed, according to Naksuriya et al., [53]. Nevertheless, in many studies, CUR has been loaded onto electrospun nanofibers, and none of them reported a loss of bioactivity and/or degradation of the aforementioned compound [10,11,13,14,34].

### 4.3. Hydrogen Nuclear Magnetic Resonance (^1^H NMR)

Appendix A show the ^1^H NMR spectrum of the curcumin control in CD_3_OD and DMSOd_6_; as can be observed, although the spectra are similar (Appendix A), the signal at 4.52 ppm assigned to the methoxy group in curcumin can be seen in CD_3_OD spectra (Appendix A), but not in the DMSOd_6_ one (Appendix A). Likewise, the aromatic signals at 7.93 and 8.13 ppm assigned to the phenyl substituted ring of curcumin can only be observed in the DMSOd_6_ [54,55]. In related work, the ^1^H NMR spectra of pure curcumin were also obtained in both CD_3_OD and DMSOd_6_, with no significant difference except for the disappearance of proton H after several days in solution. In this case, the disappearance of proton NMR signals could be attributed to a deuteration process, which is common when methanol-d_4_ is used as solvent [54] (Appendix A). Such discrepancies can be mainly attributed to the non-commercial source of curcumin; thus, the yellow curcumin leaves displayed different solubility in different solvents. Furthermore, the amount of active compound varies from leaf to leaf, which is why the spectrum with the pure substance differs.

Appendix A shows a comparison between the CUR control in both deuterated solvents, PCL, and PCL-CUR4, as before no appreciable CUR signals were observed in the sample even at a concentration of 27 µM. A higher number of scans or concentration of CUR is necessary for detection by ^1^H NMR.

### 4.4. Thermogravimetric Analysis (TGA)

Solvent evaporation can be seen as reported in the literature [56,57]. For all samples, 10% of weight loss is observed in the range of 319–379 °C. Nevertheless, the boiling point of chloroform is 61 °C [58] and ethanol is 78 °C [59], which are the solvents used for the preparation of the fibers; hence, these solvent evaporations were not visible in the TG graph. Still, the range of temperature detected at 10% weight loss of the samples corresponds to the evaporation of PCL, which is around 380 °C [50]. Solvent evaporation was not seen in the TGA graph of the samples, since both of the solvents employed in fiber preparation (CHCl_3_ and ethanol) have low boiling points. It was reported that the weight loss of PCL was found to be around 88% between 376 and 480 °C [51], which correspond to our results.

In the case of CUR, the initial weight loss that occurs below 130 °C in all samples can be related to water evaporation. The TG thermogram of CUR shows that the main decomposition started at around 270 °C and continued up to 430 °C [10].

### 4.5. Differential Scanning Calorimetry (DSC)

Schroeter et al. reported that the glass transition (T_g_) of PCL at –60 °C was not visible in the DSC graph. Moreover, the melting point of PCL is reported to be between 59–64 °C depending on its molecular weight [60]. For example, the melting point of PCL (70,000 g/mol MW) electrospun fibers is reported at 60 °C [50,61], while the T_m_ of PCL (80,000 g/mol MW) electrospun fibers is reported at 59 °C [52]. The T_m_ of our PCL fibers is 63 °C and the T_m_ of our PCL/CUR vary between 62–65 °C; it can be seen clearly how the T_m_ increase when the CUR concentration is higher, and these temperatures are slightly higher compared to the above-reported T_m_ of PCL. Fortunately, for bioreactor purposes, *S. aureus, E. coli,* and *P. aeruginosa* grow at 37 °C [62]. 

PCL is a semi-crystalline degradable polymer with sufficient mechanical strength and thermal stability [60]. The T_d_ of PCL (80,000 g/mol MW) fibers are reported to be about 376–380 °C [51,63]. Gautam et al. reported PCL degradation around 432 °C [51], with complete degradation around 460 °C, which is consistent with our results. 

### 4.6. Bacterial Growth Assay

Our observations are in agreement with several reports. Ramírez-Cedillo et al. established that PCL fibers without any treatment do not provoke the inhibition of *S. aureus* [52]. In another study, Gram-negative *P. aeruginosa* and Gram-positive *S. epidermidis*, bacterial species displayed uninhibited suspended growth in medium exposed to PCL scaffolds after 6 h at 37 °C [33]; likewise, *P. aeruginosa* proliferated rapidly while also secreting extracellular polymeric substances (EPSs), with 6 h in broth exposed to PCL scaffolds, indicating that the initial hours after inoculation are critical for inhibiting biofilm formation [33]. Finally, Adeli-sardou et al. reported that PCL/GEL fiber had no effect on biofilm-producing bacteria *S. aureus, P. mirabilis*, Methicillin-resistant *Staphylococcus aureus (MRSA)* and *P. aeruginosa* [64]. Thus, the latter and our results confirm that PCL scaffolds do not hinder the growth of Gram-positive and Gram-negative bacterial strains, and therefore they can be an excellent option to study bacterial biofilm formation.

Several studies shed light on the bioactive properties of CUR which provoke membrane damage in all the tested microorganisms [65]. With concentrations of 25, 50, and 100 µM of CUR I, and after 30 min of exposition, between ~10–20% of the cell viability was affected in most of the tested strains. CUR I at the lowest concentration of 25 µM after 120 min does not decrease cell viability more than ~20%, with the exception of *S. aureus*, which showed high susceptibility to CUR [65]. 

Researchers explained that the better protection of Gram-negative bacteria against antimicrobials is due to the specific structure of their cell walls. Lipopolysaccharides of the outer cell envelope represent the outermost permeability barrier for a variety of antimicrobial compounds, responsible for the unusually slow influx of lipophilic solutes in Gram-negative bacteria. In contrast, porin proteins embedded in the outer membrane represent the main channels for solute entry into the cells of Gram-negative bacteria [43].

Comparing the obtained results between PCL/CUR fibers and films, PCL/CUR fibers significantly increased the growth (ANOVA P < 0.05) of *Escherichia coli* and *Staphylococcus aureus* at all times (at all incubation times). For *Pseudomona aeruginosa*, growth occurred only after 24 h. PCL/CUR films, on the other hand, delay the bacterial replication rate of *Escherichia coli* and do not significantly affect the growth of *Pseudomona aeruginosa* and *Staphylococcus aureus*. The enhancing properties of the bacterial growth are attributed to the tridimensional structure that the fibers offer and the higher surface area contact that these fibrous scaffolds possess [17]. Cells did not only grow over the surface of the fibrous mat, but the porosity of these mats led to the penetration of bacterial cells into the scaffolds. This phenomenon can be clearly seen in Figure 8 and Appendix A. These affirmations can also be corroborated by comparing the cell size of each bacteria; *Escherichia coli* is a typical Gram-negative rod bacterium with cylinder dimensions of 1.0–2.0 µm in length and a radius of about 0.5 µm [66]. *Pseudomona aeruginosa* has been reported to have a size around ~10 μm [67], about 0.1 to 0.25 μm in width, and roughly 1 to 1.5 μm in length for *Staphylococcus aureus* [68]. PCL fibers created with standard electrospinning technique parameters reported fibers mats with pore sizes between 10–45 mm [69]. These data support the idea of bacterial penetration into the tridimensional structure of the PCL/CUR fibrous scaffolds, where bacterial cells proliferate inside the mat; consequently, after exposure to CUR, bacterial cells stressed and synthesized the extracellular molecules producing biofilms.

### 4.7. Biofilm Formation Study

The aim of incorporating a natural bactericidal component like CUR [70] into the inert polymeric nanofibers of PCL is to stimulate the bacterial biofilm by provoking cellular stress in the tested bacteria [6]. 

Likewise, Chu et al. proposed that mechanical stress (which is one of the cellular stresses that promotes biofilm formation) could initially emerge from the physical stress, accompanying colony confinement, within micro-cavities or hydrogel environments reminiscent of the cytosol of host cells [6]. Biofilm formation can then be enhanced by a nutrient access-modulated feedback loop, in which biofilm matrix deposition can be particularly high in areas of increased mechanical and biological stress, with the deposited matrix further enhancing the stress levels. This feedback regulation can lead to adaptive and diverse biofilm formation guided by environmental stresses. 

Hence, our PCL/CUR tridimensional scaffolds mean to generate the above mechanical stress to bacterial cells, whilst the antimicrobial effect of the CUR can increase the reactive oxygen species (ROS) and inhibit electron transport [44]. In some bacteria, like *E. coli* and *P. aeruginosa*, bacterial motility is important for cellular growth [71]; therefore, another strategy employed is electrospinning. The electrospun nanofibers will cancel the motility the bacterial cells need to replicate, thus leading to cellular stress that elicits biofilm formation.

Again, under stressful conditions, bacterial cells can actively seek out protective, spatially isolated niches, such as micro-cavities in the complex mechanical microenvironments or the cytosolic compartments of host cells. There, they can be shielded from adverse effects of the environment and grow to high densities. The resultant tight cell packing leads to various forms of collective behavior, shaped by both biological responses and mechanical effects [6].

Research in biofilm production in a controlled manner owes its interest to the potential production of interesting metabolites. Biofilms allow sessile cells to live in a coordinated, more permanent manner that favors their proliferation. The biofilm itself is a surrounding matrix that mostly is composed of exopolysaccharides that allows their attachment to inert (e.g., rocks, glass, plastic) or organic (e.g., skin, cuticle, mucosa) surfaces [5].

*Pseudomonas* strains could be considered for future use in bioremediation of contaminated water sources with oil spills. However, further studies are needed to evaluate the potential of the isolated strains to degrade hydrocarbons in situ in natural environmental conditions [8].

*Escherichia coli* is one of the bacteria with which biofilm formation has been studied in detail, and it is especially appreciated for biotechnology applications because of its genetic amenability [72]. Amongst the industrial products that can be produced by *E. coli* are lycopene [73] and succinic acid [74], among others.

Finally, *Staphylococcus aureus* and other pathogenic bacteria have been tested as heavy-metal-tolerant bacteria for the removal of Cadmium, Chrome, Nickel and Zinc, in petroleum refinery effluents [75]. Likewise, other uses include oil bioremediation in vat dyes of textile effluents [76].

## 5. Conclusions

The study of biofilm production has gained interest in two ways: first, biofilms are one of the major problems facing biomedical applications, since their presence affects a great number of medical devices. Moreover, most of the pathogenic bacteria are able to protect themselves thanks to biofilm production. On the other hand, biofilms of specific bacteria can produce metabolites that can be extracted and used in different areas of industry. *Staphylococcus aureus*, *Escherichia coli*, and *Pseudomona aeruginosa* are interesting in both ways. Hence, is important to control and manipulate biofilm formation for its study and for a better understanding of how to design standardized methods to avoid biofilm presence in medical devices or harvest bacteria biofilms for use in biotechnological approaches. In this study, we use PCL/CUR to promote biofilm formation over a controlled surface, and nanofibers successfully promoted biofilm formation in the three tested bacteria strains thanks to the strategy of using a natural bactericide like CUR as cellular stressing stimuli. In the characterization of the scaffolds, the purpose of the FTIR and ^1^H NMR studies was to demonstrate the presence of CUR in the fibers, but unfortunately, no CUR signals were able to appear due to the low concentration of CUR in the samples. The small concentration of CUR in the samples was necessary due to the objective of this work, which is the use of CUR as stress stimuli to promote biofilm formation rather than an antimicrobial agent. If higher concentrations of CUR were used, the signals for the CUR could be detected by FTIR and ^1^H NMR, but this will provoke the CUR to work as an antibacterial agent in the fibrous scaffolds, which is undesired for this work. Meanwhile, thermogravimetric analysis (TGA and DSC) showed the great thermal stability of the fibers, which will be useful for bioreactor systems. SEM micrography shows homogeneous fibers with similar sizes to those reported in the literature. Finally, this work demonstrated that no statistical difference was observed in the four different formulations of CUR. This means that it can be decided whether to use the highest or lowest concentration of CUR for the preparation of CUR loaded polymeric fibers for biotechnological applications, anticipating similar results. Nevertheless, PCL/CUR3 fibers are the most promising system. Moreover, it was demonstrated how the use of the PCL/CUR fibrous scaffolds significantly improved the growth of bacteria compared to the PCL/CUR films, and thus, effective for biofilm formation. Further studies will be performed to demonstrate the efficacy of metabolite production in a controlled electrospun mat and its extraction for biotechnological purposes.

## Figures and Tables

**Figure 1 materials-13-05556-f001:**
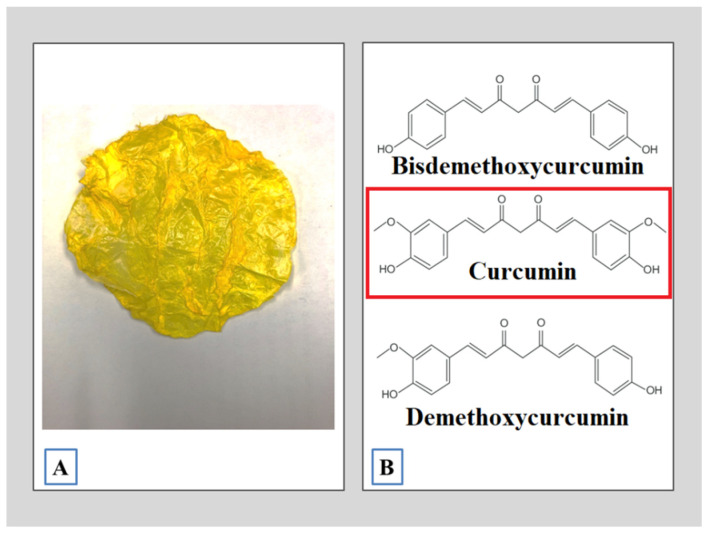
*Curcuma longa L*. (**A**) The curcuma leaf; (**B**) the main bioactive compounds present in CUR.

**Figure 2 materials-13-05556-f002:**
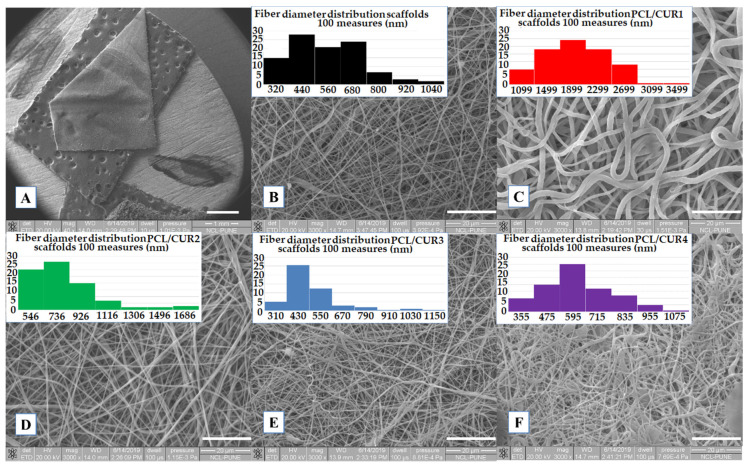
SEM images of PCL/CUR scaffolds. (**A**) General view of the membranes (1×), (**B**) PCLc, (**C**) PCL/CUR1, (**D**) PCL/CUR2, (**E**) PCL/CUR3, (**F**) PCL/CUR4. Images B–F have 3000× of amplification.

**Figure 3 materials-13-05556-f003:**
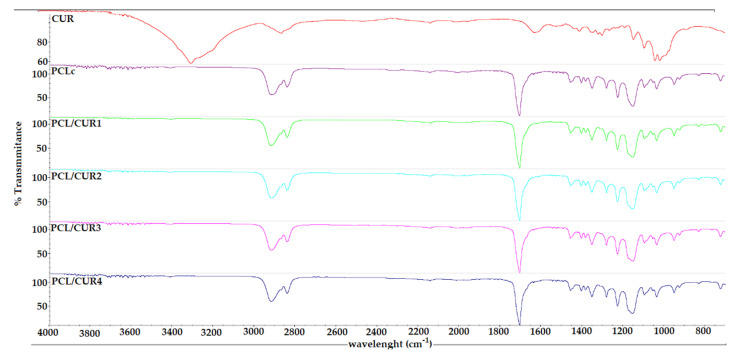
The FTIR spectra of PCL/CUR nanofibers.

**Figure 4 materials-13-05556-f004:**
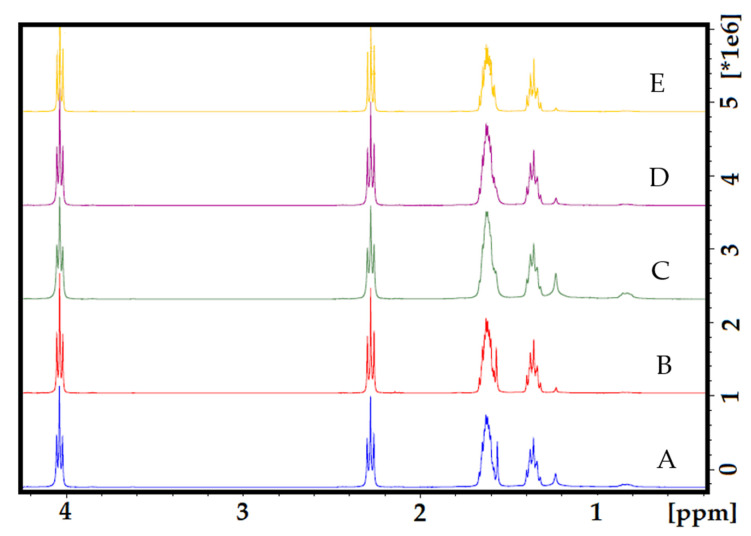
The stack of ^1^H NMR spectra of electrospun nanofibers. (**A**) PCL control; (**B**) PCL-CUR1; (**C**) PCL-CUR2; (**D**) PCL-CUR3; (**E**) PCL-CUR4.

**Figure 5 materials-13-05556-f005:**
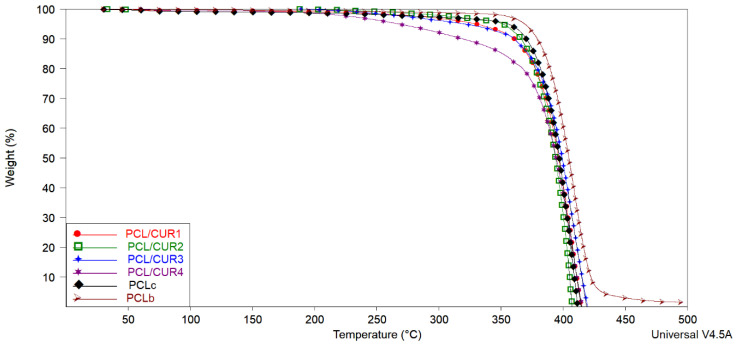
The TGA spectra of PCL/CUR scaffolds. PCL: poly (caprolactone). CUR: Curcumin. PCLc: PCL control fibers. PCLb: PCL beads.

**Figure 6 materials-13-05556-f006:**
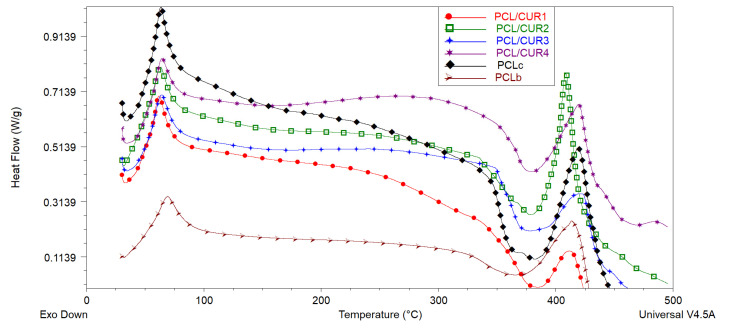
The DSC spectra of PCL/CUR scaffolds.

**Figure 7 materials-13-05556-f007:**
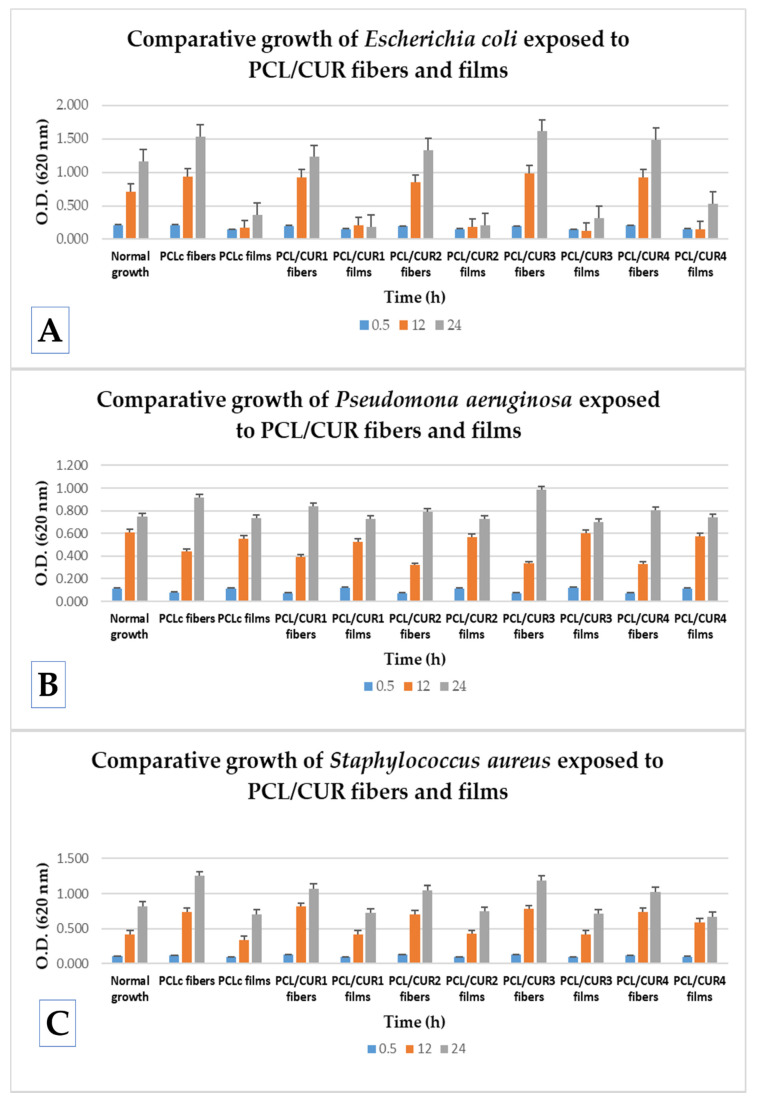
A comparative study of bacterial growth of exposed bacteria to PCL/CUR fibers and PCL/CUR films. (**A**) *Escherichia coli*; (**B**) *Pseudomona aeruginosa*; (**C**) *Staphylococcus aureus*.

**Figure 8 materials-13-05556-f008:**
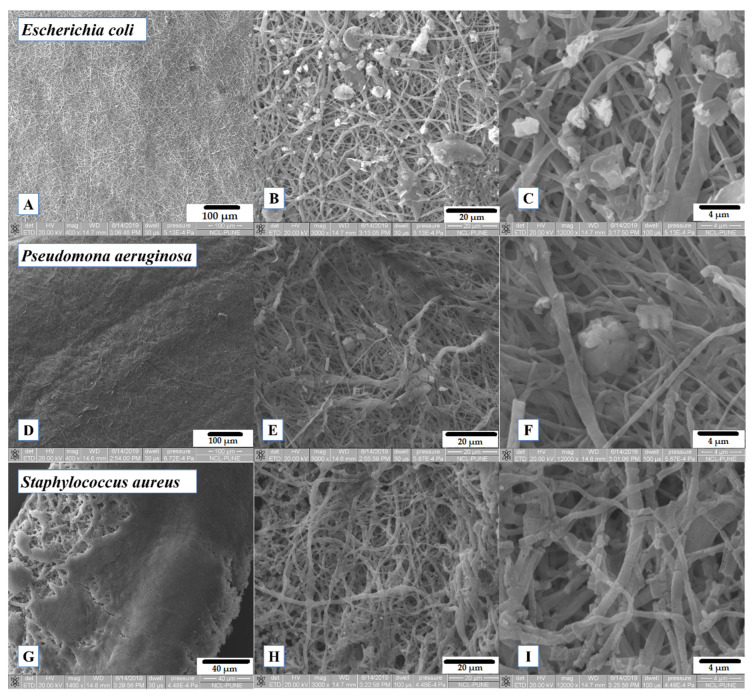
SEM images of exposed PCL/CUR4 scaffolds with bacteria after 12 h of incubation. (**A**) *Escherichia coli* on PCL/CUR4 fibers (400×); (**B**) *Escherichia coli* on PCL/CUR4 fibers (3000×); (**C**) *Escherichia coli* on PCL/CUR4 fibers (12,000×); (**D**) *Pseudomonas aeruginosa* on PCL/CUR4 fibers (400×); (**E**) *Pseudomona aeruginosa* on PCL/CUR4 fibers (3000×); (**F**) *Pseudomona aeruginosa* on PCL/CUR4 fibers (12,000×); (**G**) *Staphylococcus aureus* on PCL/CUR4 fibers (1400×); (**H**) *Staphylococcus aureus* on PCL/CUR4 fibers at (3000×); (**I**) *Staphylococcus aureus* on PCL/CUR4 fibers (12,000×).

**Table 1 materials-13-05556-t001:** Electrospun polymeric nanofibers loaded with CUR.

Electrospun Polymeric Nanofibers	Application	Reference
Almond gum/poly (vinyl alcohol) (PVA) composite	Food and pharmaceutical industries	[10]
Poly (urethane) (PU)	Antibacterial activity	[11]
Cellulose acetate phthalate (CAP)	Local skin disorders (acne and various types of wounds)	[12]
Poly (3-hydroxy butyric acid-co-3-hydroxy valeric acid) (PHBV)	Wound-dressing	[13]
Poly (ethylene oxide) (PEO) and hydroxypropyl methylcellulose (HPMC)	Drug delivery systems	[14]
Poly (lactic acid) (PLA)/cyclodextrin (CD)	Antioxidant activity	[15]

**Table 2 materials-13-05556-t002:** PCL/CUR solutions prepared for electrospinning.

Sample	CUR (%) *v/v*	Final CUR Concentration	Solution	PCL/CUR Proportion
PCLc	0	0 µM	2000 µL PCL	--
PCL/CUR1	2	5 µM	1800 µL PCL + 200 µL CUR	1.8:0.2
PCL/CUR2	2.5	7 µM	1600 µL PCL + 400 µL CUR	1.6:0.4
PCL/CUR3	5	14 µM	1200 µL PCL + 800 µL CUR	1.2:0.8
PCL/CUR4	10	27 µM	1000 µL PCL + 1000 µL CUR	1:1

Note: All PCL solutions were at 10%.

**Table 3 materials-13-05556-t003:** The average fiber diameter and % porosity of PCL/CUR nanofibers.

Sample	Average Fiber Diameter ± SD (nm)	Frequency/Fiber Diameter (nm)	Superficial Porosity (%)	3D Porosity (%)
PCLc	500 ± 165	320–440	50	13
PCL/CUR1	1734 ± 525	1499–1899	48	10
PCL/CUR2	709 ± 254	546–736	51	12
PCL/CUR3	441 ± 154	310–430	44	8
PCL/CUR4	557 ± 161	475–595	36	6

SD: Standard deviation.

**Table 4 materials-13-05556-t004:** % weight loss vs. temperature increment in PCL/CUR scaffolds.

Sample	Weight Loss (%)	Temperature °C
PCL/CUR1	10	361
	50	397
	100	415
PCL/CUR2	10	366
	50	394
	100	407
PCL/CUR3	10	361
	50	399
	100	420
PCL/CUR4	10	319
	50	394
	100	415
PCLc	10	370
	50	397
	100	412
PCLb	10	379
	50	405
	100	496

PCLc: PCL fibers control; PCLb: PCL beads.

**Table 5 materials-13-05556-t005:** The thermal behavior of PCL/CUR nanofibers.

Sample	Tm °C	Td °C	T_max_ °C
PCLb	68	367	414
PCLc	63	381	420
PCL/CUR1	62	386	411
PCL/CUR2	63	377	409
PCL/CUR3	64	377	423
PCL/CUR4	65	380	420

Tm: melting temperature; Td: degradation temperature; PCLc: PCL fibers control; PCLb: PCL beads.

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
