# Peer review of "Bacterial Biofilm Formation Using PCL/Curcumin Electrospun Fibers and Its Potential Use for Biotechnological Applications"

_materials, 2020, doi:10.3390/ma13235556_

Round 1

Reviewer 1 Report

Interesting and well-written. This work is also innovative, because usually most of the existing bibliography about it is oriented on eradication and inhibition of biofilm formation, while here the bacterial biofilm is treated as a potential resource for various biotechnological uses.
However, I would have to ask for some clarifications and to suggest some minor revisions:

Introduction section:

  • In the first part of the introduction, when the authors describe bacterial biofilms as important medical problems, they cannot ignore the addition, with appropriate bibliographical references, that one of the most important characteristics that contributes to the severity of biofilm infections is the their peculiar resistance to antibiotics.
  • The authors should add, to the already mentioned uses of electrospun fibers, also the very current one of skin regeneration (Azimi et al., 2020),
    moreover there are more recent works than those cited, such as  Azimi, Sorayani Bafqi MS et al. (2020) for tissue engineering applications.

Material and methods section:

  • At the line 126, it's not "Muller- Hinton" but "MUELLER-Hinton"
  • In the paragraph 2.5, only the materials with clean medium should have been included among the negative controls in order to verify that the sterilization took place correctly. If not, there could be some false positives.
  • Although Mueller-Hinton is widely used in antimicrobial activity assays, in biofilm assays the use of a medium that is not optimal for the various strains used (indicated in the ATCC sheet relating to each of them), could interfere with the biofilm formation ability. Was it verified that there was no difference in the result of assay? Additionally, biofilm formation assays for the bacteria used in this work are usually conducted for 24 hours. How come in this case we only proceeded for 12 hours? Probably with a longer time, even more certain results would be obtained.
  • The biofilm formation assay should have been carried out in parallel also in polystyrene plates (Fusco et al., 2017) in the absence of the scaffolds, in order to evaluate a possible difference between the production speed and the quantity of biofilm produced with and without scaffold.

Results section:

  • In the paragraph 3.6, Fig. 7 shows the graphs with the relative growth curves of the different strains under different conditions. Since the PCL fibers have been conjugated with curcumin at various concentrations, why has pure curcumin been tested at only one concentration (which is also not indicated)? It would have been more correct to test all the percentages used with PCL fibers to assess whether the loss of bacterial growth inhibition of curcumin was due to the preparation technique
  • Line 260: the sentence needs to be better clarified.
  • Line 276-277: the sentence needs to be better clarified.
  • Figure 8: why is the data of PCL / CUR4 only shown? It is not always certain that the best effects are obtained with the highest concentrations.

Discussion section:

  • lane 381: the sentence needs to be better clarified.

Author Response

Dear Reviewer

On behalf of all the authors, we want to thank you and the other reviewers for the time in revising this manuscript, we are grateful for the opportunity to improve its quality. Below you will find the responses and corrections of each comment and suggestions of the 3 reviewers. We appreciated the quality of the reviewer’s comments and we strongly believe that the quality of this manuscript has been improved, hopefully being enough for publication in this important journal. Some of the highlights of this manuscript are as follows:

  • PCL/CUR scaffolds successfully improved the formation of Escherichia coli, Staphylococcus aureus and Pseudomona aeruginosa biofilms.
  • PCL/CUR fibers possess high temperature resilience with degradation temperatures over > 350 oC adequate for bioreactor purposes
  • Is important to control and manipulated biofilm formation, for its study and to better understanding to design of standardized methods to avoid biofilm presence in medical devices, or to use harvested bacteria biofilms for use in biotechnological approaches
  • PCL fibers successfully improved biofilms formation of the three tested bacteria, thanks to the strategy of using a natural bactericide CUR as cellular stressing stimuli.
  • In the comparison study between fibers and films, the results showed how PCL/CUR fibers substantially enhanced the bacterial proliferation compared to the PCL/CUR films.

Note: You can find your comments and responses below attached with images

Dr. Luis Jesús Villarreal Gómez

__________________________________________________

Reviewer 1

Interesting and well-written. This work is also innovative, because usually most of the existing bibliography about it is oriented on eradication and inhibition of biofilm formation, while here the bacterial biofilm is treated as a potential resource for various biotechnological uses.
However, I would have to ask for some clarifications and to suggest some minor revisions:

 Introduction section:

In the first part of the introduction, when the authors describe bacterial biofilms as important medical problems, they cannot ignore the addition, with appropriate bibliographical references, that one of the most important characteristics that contributes to the severity of biofilm infections is their peculiar resistance to antibiotics.

Response: Thank you for your comment; we add the following sentence “Most importantly, one of the most predominant characteristics that contributes to the severity of biofilm infections is their peculiar resistance to antibiotics [2-4].”  The corresponding references that support the idea were included:

  1. Sharma, D.; Misba, L.; Khan, A. U. Antibiotics versus Biofilm: An Emerging Battleground in Microbial Communities. Antimicrob. Resist. Infect. Control 2019, 8 (1), 1–10. https://doi.org/10.1186/s13756-019-0533-3.
  2. Høiby, N.; Bjarnsholt, T.; Givskov, M.; Molin, S.; Ciofu, O. Antibiotic Resistance of Bacterial Biofilms. Int. J. Antimicrob. Agents 2010, 35 (4), 322–332. https://doi.org/https://doi.org/10.1016/j.ijantimicag.2009.12.011.
  3. Dincer, S.; Masume, F.; Anil, D. Antibiotic Resistance in Biofilm. In Bacterial Biofilms; IntechOpen, 2020; p 153.

Location: Page 2, lines 42-43

  • The authors should add, to the already mentioned uses of electrospun fibers, also the very current one of skin regeneration (Azimi et al., 2020),
    moreover there are more recent works than those cited, such as Azimi, Sorayani Bafqi MS et al. (2020) for tissue engineering applications.

Response: Thank you for your comment, we add the following sentence “Electrospun nanofibers have a myriad of applications, for instance in tissue engineering [16, 17], such as skin regeneration [18], wound dressing [19], cartilage regeneration [20], bone regeneration [21], amidst others. Applications also include sensors [22, 23], drug delivery systems [24, 25], filters [26, 27], and solar cells [28-29], amongst others”. We included the following references that support the latter:

  1. Azimi, B.; Thomas, L.; Fusco, A.; Kalaoglu-Altan, O. I.; Basnett, P.; Cinelli, P.; De Clerck, K.; Roy, I.; Donnarumma, G.; Coltelli, M.-B.; Danti, S.; Lazzeri, A. Electrosprayed Chitin Nanofibril/Electrospun Polyhydroxyalkanoate Fiber Mesh as Functional Nonwoven for Skin Application. J. Funct. Biomater. 2020, 11 (3), 62. https://doi.org/10.3390/jfb11030062.
  2. Álvarez-Suárez, A. S.; Dastager, S. G.; Bogdanchikova, N.; Grande, D.; Pestryakov, A.; García-Ramos, J. C.; Pérez-González, G. L.; Juárez-Moreno, K.; Toledano-Magaña, Y.; Smolentseva, E.; Paz-González, J. A.; Popova, T.; Rachkovskaya, L.; Nimaev, V.; Kotlyarova, A.; Korolev, M.; Letyagin, A.; Villarreal-Gómez, L. J. Electrospun Fibers and Sorbents as a Possible Basis for Effective Composite Wound Dressings. Micromachines 2020, 11 (4), 441. https://doi.org/10.3390/mi11040441.
  3. Yilmaz, E. N.; Zeugolis, D. I. Electrospun Polymers in Cartilage Engineering—State of Play. Front. Bioeng. Biotechnol. 2020, 8 (February), 1–17. https://doi.org/10.3389/fbioe.2020.00077.
  4. Aoki, K.; Haniu, H.; Kim, Y. A.; Saito, N. The Use of Electrospun Organic and Carbon Nanofibers in Bone Regeneration. Nanomaterials 2020, 10 (3). https://doi.org/10.3390/nano10030562.

Location: Page 3, lines 66-69

Material and methods section:

  • At the line 126, it's not "Muller- Hinton" but "MUELLER-Hinton"

Response: thank you for that observation we have corrected and revised the whole document for other similar errors.

Location: Page 5, line 148

  • In the paragraph 2.5, only the materials with clean medium should have been included among the negative controls in order to verify that the sterilization took place correctly. If not, there could be some false positives.

Response: Thank you for that observation, effectively we did that, but we missed to add this detailed information in the manuscript. We corrected this section as follows: “As a negative control, clean media, and clean media with fibers and films were used.  As a positive control, 150 μL of medium and 50 μL of each strain were placed without fibers or films.

Location: Page 5, lines 152-154

  • Although Mueller-Hinton is widely used in antimicrobial activity assays, in biofilm assays the use of a medium that is not optimal for the various strains used (indicated in the ATCC sheet relating to each of them), could interfere with the biofilm formation ability. Was it verified that there was no difference in the result of assay? Additionally, biofilm formation assays for the bacteria used in this work are usually conducted for 24 hours. How come in this case we only proceeded for 12 hours? Probably with a longer time, even more certain results would be obtained.

Response: Thanks for these two observations and we are in agreement with you. We added the following ideas to the text “Despite the fact, that several research groups have studied the biofilm formation for 24 h [46, 47]; several studies have employed shorter incubation times [48, 49]. In the present study, the objective is to observe how bacterial colonies attach to the surface of the fibers and proliferate inside the tridimensional structure of the scaffolds creating the biofilms. However, for future work, we intend to study the formation of bacterial biofilms after 24 h and evaluate its effectivity in several biotechnological application studies.” And the following references that support the idea were added:

  1. Moormeier, D. E.; Bayles, K. W. Staphylococcus Aureus Biofilm: A Complex Developmental Organism Graphical Abstract HHS Public Access. Mol Microbiol 2017, 104 (3), 365–376. https://doi.org/10.1111/mmi.13634.Staphylococcus.
  2. Ripa, R.; Shen, A. Q.; Funari, R. Detecting Escherichia Coli Biofilm Development Stages on Gold and Titanium by Quartz Crystal Microbalance. ACS Omega 2020, 5 (5), 2295–2302. https://doi.org/10.1021/acsomega.9b03540.
  3. Flament-Simon, S. C.; Duprilot, M.; Mayer, N.; García, V.; Alonso, M. P.; Blanco, J.; Nicolas-Chanoine, M. H. Association between Kinetics of Early Biofilm Formation and Clonal Lineage in Escherichia Coli. Front. Microbiol. 2019, 10 (MAY), 1–11. https://doi.org/10.3389/fmicb.2019.01183.
  4. Rasamiravaka, T.; Labtani, Q.; Duez, P.; Jaziri, and Mondher El. The Formation of Biofilms by Pseudomonas Aeruginosa: A Review of the Natural and Synthetic Compounds Interfering with Control Mechanisms. Biomed Res. Int. 2015, 2015, 1–17.

Location: Page 14, lines 396-401

In the case of the use of Mueller Hilton media in the biofilm formation test; we agreed that more optimal media always can be found. We used this media in order to avoid increasing the number of variables in bacterial growth, and thus compare both assays (bacterial growth assay and biofilm formation study). We also add this following reference to the methodology section, where the group used Mueller Hilton for Biofilm formation of Pseudomonas aeruginosa and Staphylococcus aureus.

  1. Masadeh, M. M.; Mhaidat, N. M.; Alzoubi, K. H.; Hussein, E. I.; Al-Trad, E. I. In vitro Determination of the Antibiotic Susceptibility of Biofilm-Forming Pseudomonas aeruginosa and Staphylococcus aureus: Possible Role of Proteolytic Activity and Membrane Lipopolysaccharide. Infect. Drug Resist. 2013, 6, 27–32. https://doi.org/10.2147/IDR.S41501.

Location: Reference 36, Page 5, line 162

  • The biofilm formation assay should have been carried out in parallel also in polystyrene plates (Fusco et al., 2017) in the absence of the scaffolds, in order to evaluate a possible difference between the production speed and the quantity of biofilm produced with and without scaffold.

Response: Thank you for that observation. Effectively, normal growth of bacteria was determined by the growth of bacteria on polystyrene plates in the absence of scaffolds. We added the following sentence

“As a positive control, 150 μL of medium and 50 μL of each strain were placed without fibers or films.”

Location: Page 5, lines 152-153

For comparison purposes, we repeated the bacterial growth assay using the fibers and casting films, with same composition of PCL/CUR; including controls without any sample. The biofilm formation assay as you can see is qualitative and we don’t quantify the amount of biofilm produced. Nevertheless, we quantified the bacteria proliferation with the bacterial growth assay.

We added the following:

“2.4. Preparation of PCL/CUR cast films

PCL and PCL/CUR casting films were prepared using PCL and PCL/CUR solutions obtained in section 2.2. The membranes were obtained by casting 1.5 mL of each solution in a circular mold (diameter 2.5 cm), followed by the evaporation of solvent at room temperature according to Brianezi, et al., [35].”

  1. Brianezi, S. F. S.; Castro, K. C.; Piazza, R. D.; do Socorro Fernandes Melo, M.; Pereira, R. M.; Marques, R. F. C.; Campos, M. G. N. Preparation and Characterization of Chitosan/MPEG-PCL Blended Membranes for Wound Dressing and Controlled Gentamicin Release. Mater. Res. 2018, 21 (6), e20170951. https://doi.org/10.1590/1980-5373-mr-2017-0951.

Location: Page 4, lines 108-113, reference 35

“Nanofibers mats and cast films samples were cut with a 0.5 cm diameter using a drill and sterilized on both sides with UV-light radiation for 15 min, and placed at the bottom of a 96 plate-wells.”

Location:  Page 5, lines 144-146

 “As a negative control, clean media, clean media with fibers and films were used. As a positive control, 150 μL of medium and 50 μL of each strain were placed without fibers or films.”

Location: Page 5, lines 152-154

“Bacterial growth studies are important to determine whether PCLc, and PCL/CUR fibers better increased bacterial proliferation, compared to PCLc and PCL/CUR films”

Location: Page 11, lines 278, 279

In order to better explain the obtained results; the percentage of proliferation rate was calculated, taking into account the normal growth of each bacteria (bacterial suspension without fibers and films). Where O.D. is the optical density obtained at 620 nm after the exposed time.

                   (1)

Comparing the efficiency in the increment of the bacterial proliferation between PCL/CUR fiber and PCL/CUR films; for Escherichia coli, after 30 min of incubation both fibers and films delay the normal growth, ~7% for fibers and ~30% for films, respectively. Then, after the 12 h, around ~30% of improved growth was observed in the presence of fibers and a huge reduction of growth of about ~ 73% less was elicited from the films. The same behavior was observed after 24 h of incubation.

Hence, for E. coli, PCL/CUR nanofibers clearly increment the proliferation rate and PCL/CUR films alter the normal growth. When comparing between each individual formulation of films and fibers; the most efficient fibrous scaffold was PCL/CUR3, which increased the growth about 37% higher than the normal growth in 12 and 24 h; following with the PCLc fibers which enhanced growth about a 32% at 24 h. The less interesting fibrous scaffold was PCL/CUR1 with just 5% of more growth than the control at 24 h.

No clear difference can be seen between PCL/CUR fibers, with respect to its bioactive effect, when the concentration of CUR was varied. This last result is valuable to assess the appropriate concentration of CUR, depending on the application and expected results. necessity to use certain concentration of CUR into the fibers to obtain the expected results.

 In the case of Pseudomona aeruginosa growth, a different bacterial behavior was observed compare to the latter results. In general, PCL/CUR films didn’t affect bacterial growth at any time interval tested (ANOVA P < 0.05). On the contrary, in contrast PCL/CUR fibers delay the growth by ~52% after the first 12 h; but after 24 h all PCL/CUR fibers increase the proliferation rate ~15%, this can be due to an adaptation period. At 24 h, PCL/CUR3 presented the best results with an increment of (~31%), followed by the PCLc fibers with 22% of enhanced growth with respect to the control. As with the PCL/CUR films, PCL/CUR2, PCL/CUR4 do not significatively affect the normal growth of P. aeruginosa.

Finally, PCL/CUR fibers efficiently promoted the growth of Staphylococcus aureus compared to PCL/CUR films. At 0.5 h, PCL/CUR fibers promoted ~16% the proliferation rate, and PCL/CUR films reduced the growth by ~11%. Then, at 12 h, PCL/CUR fibers enhanced about ~80% the cellular replication whilst PCL/CUR films just enhanced ~5%. Lastly, after 24 h, PCL/CUR fibers promoted the growth by ~36% and PCL/CUR films reduced the normal growth by ~14%. The PCL/CUR fibers with the best results was PCL/CUR1 with ~94% of enhanced growth at 12 h, followed by the PCL/CUR3 fibers, which promoted the replication of S. aureus by ~86% at 12 h.

Hence, it can be demonstrated that PCL/CUR fibers are efficient systems that promoted the growth of all the tested bacterial, particularly these fibrous scaffolds were more effective in promoting growth in Staphylococcus aureus and Escherichia coli.  Bacteria Exposed to PCL/CUR films presented different behaviors depending on the bacterial strain. From all of the PCL/CUR fibers, the system with best results was PCL/CUR3, but no statistically difference between CUR concentration was found (ANOVA P < 0.05). The behavior of exposed bacteria toward the PCL/CUR scaffolds is dependent on the bacterial strain. Nonetheless the system with the best results was PCL/CUR3, although no statistically difference between CUR concentration was found (ANOVA P < 0.05).

In the present study, to ensure the sterilization process (15 min UV irradiation for both sides), all PCL/CUR fibers and films were exposed to clean media, to observe the presence of any growth; but no changes in the O.D. was observed. Moreover, to demonstrated that different CUR concentrations have an effect on the bacteria, as well as confirm the capacity of CUR to cause cellular stress; pure CUR ethanolic extract (dried) at different concentrations (CUR1: 5 µM; CUR2: 7 µM; CUR3: 14 µM and CUR4: 27 µM) and pure CUR leaf were tested in presence of the three bacterial strains. It can be observed that the unprocessed CUR leaf does not significantly affect the normal growth of each bacteria (ANOVA P < 0.05); but, dried CUR ethanolic extracts decreased the growth proportional to concentration, therefore, the higher the concentration of CUR employed, the higher the reduction of proliferation is, in all three bacteria at all times, albeit this decrement is not significant different (ANOVA P < 0.05) between CUR concentrations, a tendency is clearly observed (Figure S2).

Figure 7. Comparative study of bacterial growth of exposed bacteria to PCL/CUR fibers and PCL/CUR films. A) Escherichia coli. B) Pseudomona aeruginosa. C) Staphylococcus aureus.

Location: Pages 11-13; lines 285-366, Figure 7, Figure S2

Figure S2. Relative growth of exposed bacteria to different concentrations of CUR. (a) Escherichia coli. B) Pseudomona aeruginosa. C) Staphylococcus aureus

Location: Supplemental material Figure S2

“Comparing the obtained results between PCL/CUR fibers and films; PCL/CUR fibers significantly increased the growth (ANOVA P < 0.05) of Escherichia coli and Staphylococcus aureus at all times (at all incubation times). For Pseudomona aeruginosa growth occurred only after 24.  PCL/CUR films on the other hand, delay the bacterial replication rate of Escherichia coli and not significantly affect the growth of Pseudomona aeruginosa and Staphylococcus aureus. The enhancing properties of bacterial growth are attributed to the tridimensional structure that fibers offer and the higher surface-area contact that these fibrous scaffolds possess [17]. Cells did not only growth over the surface of the fibrous mat, but the porosity of these mats led to the penetration of bacterial cells into the scaffolds. This phenomenon can be clearly seen in figure 8 and figure S3. These affirmations can also be corroborated by comparing the cell size of each bacteria; Escherichia coli is a typical Gram-negative rod bacterium; with cylinder dimensions of 1.0-2.0 µm long cylinder and radius of about 0.5 µm [66]. Pseudomona aeruginosa have been reported to have a size around ~10 μm [67], and about 0.1 to 0.25 μm in width and roughly 1 to 1.5 μm in length for Staphylococcus aureus [68]. PCL fibers created with standard electrospinning technique parameters reported fibers mats with a pore size between 10–45 mm [69]. These data support the idea of bacterial penetration into the tridimensional structure of the PCL/CUR fibrous scaffolds, where bacterial cells proliferate inside the mat; consequently, after exposure to CUR, bacterial cells stressed and synthesized the extracellular molecules producing biofilms.”

Location: Page 17, 18; lines 514-531

  1. Zheng, H.; Ho, P. Y.; Jiang, M.; Tang, B.; Liu, W.; Li, D.; Yu, X.; Kleckner, N. E.; Amir, A.; Liu, C. Interrogating the Escherichia Coli Cell Cycle by Cell Dimension Perturbations. Proc. Natl. Acad. Sci. U. S. A. 2016, 113 (52), 15000–15005. https://doi.org/10.1073/pnas.1617932114.
  2. Deforet, M.; Van Ditmarsch, D.; Xavier, J. B. Cell-Size Homeostasis and the Incremental Rule in a Bacterial Pathogen. Biophys. J. 2015, 109 (3), 521–528. https://doi.org/10.1016/j.bpj.2015.07.002.
  3. Monteiro, J. M.; Fernandes, P. B.; Vaz, F.; Pereira, A. R.; Tavares, A. C.; Ferreira, M. T.; Pereira, P. M.; Veiga, H.; Kuru, E.; Vannieuwenhze, M. S.; Brun, Y. V.; Filipe, S. R.; Pinho, M. G. Cell Shape Dynamics during the Staphylococcal Cell Cycle. Nat. Commun. 2015, 6 (1), 8055. https://doi.org/10.1038/ncomms9055.
  4. Rnjak-Kovacina, J.; Weiss, A. S. Increasing the Pore Size of Electrospun Scaffolds. Tissue Eng. - Part B Rev. 2011, 17 (5), 365–372. https://doi.org/10.1089/ten.teb.2011.0235.

Location: References 66-69

Finally, this work demonstrated that, no statistical difference was observed in the four different formulations of CUR. This means that, it can be decided whether to use the highest or lowest concentration of CUR for the preparation of CUR loaded polymeric fibers for biotechnological applications, anticipating similar results. Nevertheless, PCL/CUR3 fibers are the most promising system. Moreover, it was demonstrated how the use of the PCL/CUR fibrous scaffolds significantly improved the growth of bacteria, compared to the PCL/CUR films, and thus, effective for biofilm formation.

Location: Page 19, lines 591-597

Results section:

  • In the paragraph 3.6, Fig. 7 shows the graphs with the relative growth curves of the different strains under different conditions. Since the PCL fibers have been conjugated with curcumin at various concentrations, why has pure curcumin been tested at only one concentration (which is also not indicated)? It would have been more correct to test all the percentages used with PCL fibers to assess whether the loss of bacterial growth inhibition of curcumin was due to the preparation technique

Response: In order to improve the quality of the assay, we repeated the experiment taking into account the evaluation of different proportions of CUR alone and conjugated with PCL in fibers and casting films.

We added the following:

“2.4. Preparation of PCL/CUR cast films

PCL and PCL/CUR casting films were prepared using PCL and PCL/CUR solutions obtained in section 2.2. The membranes were obtained by casting 1.5 mL of each solution in a circular mold (diameter 2.5 cm), followed by the evaporation of solvent at room temperature according to Brianezi, et al., [35].”

  1. Brianezi, S. F. S.; Castro, K. C.; Piazza, R. D.; do Socorro Fernandes Melo, M.; Pereira, R. M.; Marques, R. F. C.; Campos, M. G. N. Preparation and Characterization of Chitosan/MPEG-PCL Blended Membranes for Wound Dressing and Controlled Gentamicin Release. Mater. Res. 2018, 21 (6), e20170951. https://doi.org/10.1590/1980-5373-mr-2017-0951.

Location: Page 4, lines 108-113, reference 35

“Nanofibers mats and cast films samples were cut with a 0.5 cm diameter using a drill and sterilized on both sides with UV-light radiation for 15 min, and placed at the bottom of a 96 plate-wells.”

Location:  Page 5, lines 144-146

 “As a negative control, clean media, clean media with fibers and films were used. As a positive control, 150 μL of medium and 50 μL of each strain were placed without fibers or films.”

Location: Page 5, lines 152-154

“Bacterial growth studies are important to determine whether PCLc, and PCL/CUR fibers better increased bacterial proliferation, compared to PCLc and PCL/CUR films”

Location: Page 11, lines 278, 279

In order to better explain the obtained results; the percentage of proliferation rate was calculated, taking into account the normal growth of each bacteria (bacterial suspension without fibers and films). Where O.D. is the optical density obtained at 620 nm after the exposed time.

                   (1)

Comparing the efficiency in the increment of the bacterial proliferation between PCL/CUR fiber and PCL/CUR films; for Escherichia coli, after 30 min of incubation both fibers and films delay the normal growth, ~7% for fibers and ~30% for films, respectively. Then, after the 12 h, around ~30% of improved growth was observed in the presence of fibers and a huge reduction of growth of about ~ 73% less was elicited from the films. The same behavior was observed after 24 h of incubation.

Hence, for E. coli, PCL/CUR nanofibers clearly increment the proliferation rate and PCL/CUR films alter the normal growth. When comparing between each individual formulation of films and fibers; the most efficient fibrous scaffold was PCL/CUR3, which increased the growth about 37% higher than the normal growth in 12 and 24 h; following with the PCLc fibers which enhanced growth about a 32% at 24 h. The less interesting fibrous scaffold was PCL/CUR1 with just 5% of more growth than the control at 24 h.

No clear difference can be seen between PCL/CUR fibers, with respect to its bioactive effect, when the concentration of CUR was varied. This last result is valuable to assess the appropriate concentration of CUR, depending on the application and expected results. necessity to use certain concentration of CUR into the fibers to obtain the expected results.

 In the case of Pseudomona aeruginosa growth, a different bacterial behavior was observed compare to the latter results. In general, PCL/CUR films didn’t affect bacterial growth at any time interval tested (ANOVA P < 0.05). On the contrary, in contrast PCL/CUR fibers delay the growth by ~52% after the first 12 h; but after 24 h all PCL/CUR fibers increase the proliferation rate ~15%, this can be due to an adaptation period. At 24 h, PCL/CUR3 presented the best results with an increment of (~31%), followed by the PCLc fibers with 22% of enhanced growth with respect to the control. As with the PCL/CUR films, PCL/CUR2, PCL/CUR4 do not significatively affect the normal growth of P. aeruginosa.

Finally, PCL/CUR fibers efficiently promoted the growth of Staphylococcus aureus compared to PCL/CUR films. At 0.5 h, PCL/CUR fibers promoted ~16% the proliferation rate, and PCL/CUR films reduced the growth by ~11%. Then, at 12 h, PCL/CUR fibers enhanced about ~80% the cellular replication whilst PCL/CUR films just enhanced ~5%. Lastly, after 24 h, PCL/CUR fibers promoted the growth by ~36% and PCL/CUR films reduced the normal growth by ~14%. The PCL/CUR fibers with the best results was PCL/CUR1 with ~94% of enhanced growth at 12 h, followed by the PCL/CUR3 fibers, which promoted the replication of S. aureus by ~86% at 12 h.

Hence, it can be demonstrated that PCL/CUR fibers are efficient systems that promoted the growth of all the tested bacterial, particularly these fibrous scaffolds were more effective in promoting growth in Staphylococcus aureus and Escherichia coli.  Bacteria Exposed to PCL/CUR films presented different behaviors depending on the bacterial strain. From all of the PCL/CUR fibers, the system with best results was PCL/CUR3, but no statistically difference between CUR concentration was found (ANOVA P < 0.05). The behavior of exposed bacteria toward the PCL/CUR scaffolds is dependent on the bacterial strain. Nonetheless the system with the best results was PCL/CUR3, although no statistically difference between CUR concentration was found (ANOVA P < 0.05).

In the present study, to ensure the sterilization process (15 min UV irradiation for both sides), all PCL/CUR fibers and films were exposed to clean media, to observe the presence of any growth; but no changes in the O.D. was observed. Moreover, to demonstrated that different CUR concentrations have an effect on the bacteria, as well as confirm the capacity of CUR to cause cellular stress; pure CUR ethanolic extract (dried) at different concentrations (CUR1: 5 µM; CUR2: 7 µM; CUR3: 14 µM and CUR4: 27 µM) and pure CUR leaf were tested in presence of the three bacterial strains. It can be observed that the unprocessed CUR leaf does not significantly affect the normal growth of each bacteria (ANOVA P < 0.05); but, dried CUR ethanolic extracts decreased the growth proportional to concentration, therefore, the higher the concentration of CUR employed, the higher the reduction of proliferation is, in all three bacteria at all times, albeit this decrement is not significant different (ANOVA P < 0.05) between CUR concentrations, a tendency is clearly observed (Figure S2).

Figure 7. Comparative study of bacterial growth of exposed bacteria to PCL/CUR fibers and PCL/CUR films. A) Escherichia coli. B) Pseudomona aeruginosa. C) Staphylococcus aureus.

Location: Pages 11-13; lines 285-366, Figure 7, Figure S2

Figure S2. Relative growth of exposed bacteria to different concentrations of CUR. (a) Escherichia coli. B) Pseudomona aeruginosa. C) Staphylococcus aureus

Location: Supplemental material Figure S2

“Comparing the obtained results between PCL/CUR fibers and films; PCL/CUR fibers significantly increased the growth (ANOVA P < 0.05) of Escherichia coli and Staphylococcus aureus at all times (at all incubation times). For Pseudomona aeruginosa growth occurred only after 24.  PCL/CUR films on the other hand, delay the bacterial replication rate of Escherichia coli and not significantly affect the growth of Pseudomona aeruginosa and Staphylococcus aureus. The enhancing properties of bacterial growth are attributed to the tridimensional structure that fibers offer and the higher surface-area contact that these fibrous scaffolds possess [17]. Cells did not only growth over the surface of the fibrous mat, but the porosity of these mats led to the penetration of bacterial cells into the scaffolds. This phenomenon can be clearly seen in figure 8 and figure S3. These affirmations can also be corroborated by comparing the cell size of each bacteria; Escherichia coli is a typical Gram-negative rod bacterium; with cylinder dimensions of 1.0-2.0 µm long cylinder and radius of about 0.5 µm [66]. Pseudomona aeruginosa have been reported to have a size around ~10 μm [67], and about 0.1 to 0.25 μm in width and roughly 1 to 1.5 μm in length for Staphylococcus aureus [68]. PCL fibers created with standard electrospinning technique parameters reported fibers mats with a pore size between 10–45 mm [69]. These data support the idea of bacterial penetration into the tridimensional structure of the PCL/CUR fibrous scaffolds, where bacterial cells proliferate inside the mat; consequently, after exposure to CUR, bacterial cells stressed and synthesized the extracellular molecules producing biofilms.”

Location: Page 17, 18; lines 514-531

  1. Zheng, H.; Ho, P. Y.; Jiang, M.; Tang, B.; Liu, W.; Li, D.; Yu, X.; Kleckner, N. E.; Amir, A.; Liu, C. Interrogating the Escherichia Coli Cell Cycle by Cell Dimension Perturbations. Proc. Natl. Acad. Sci. U. S. A. 2016, 113 (52), 15000–15005. https://doi.org/10.1073/pnas.1617932114.
  2. Deforet, M.; Van Ditmarsch, D.; Xavier, J. B. Cell-Size Homeostasis and the Incremental Rule in a Bacterial Pathogen. Biophys. J. 2015, 109 (3), 521–528. https://doi.org/10.1016/j.bpj.2015.07.002.
  3. Monteiro, J. M.; Fernandes, P. B.; Vaz, F.; Pereira, A. R.; Tavares, A. C.; Ferreira, M. T.; Pereira, P. M.; Veiga, H.; Kuru, E.; Vannieuwenhze, M. S.; Brun, Y. V.; Filipe, S. R.; Pinho, M. G. Cell Shape Dynamics during the Staphylococcal Cell Cycle. Nat. Commun. 2015, 6 (1), 8055. https://doi.org/10.1038/ncomms9055.
  4. Rnjak-Kovacina, J.; Weiss, A. S. Increasing the Pore Size of Electrospun Scaffolds. Tissue Eng. - Part B Rev. 2011, 17 (5), 365–372. https://doi.org/10.1089/ten.teb.2011.0235.

Location: References 66-69

Finally, this work demonstrated that, no statistical difference was observed in the four different formulations of CUR. This means that, it can be decided whether to use the highest or lowest concentration of CUR for the preparation of CUR loaded polymeric fibers for biotechnological applications, anticipating similar results. Nevertheless, PCL/CUR3 fibers are the most promising system. Moreover, it was demonstrated how the use of the PCL/CUR fibrous scaffolds significantly improved the growth of bacteria, compared to the PCL/CUR films, and thus, effective for biofilm formation.

Location: Page 19, lines 591-597

  • Line 260: the sentence needs to be better clarified. Pag 7

Response: This particular sentence was deleted and replace by “Hence, it can be demonstrated how PCL/CUR fibers are efficient systems that promoted all the three tested bacterial, been the most these fibrous scaffolds more effective in Staphylococcus aureus and Escherichia coli.  Exposed bacteria to PCL/CUR films presented different behaviors depending of the bacterial strain. From all PCL/CUR fibers, the system with best results was PCL/CUR3, but not statistically difference between CUR concentration was found (ANOVA P < 0.05).”

Location: Page 12; lines 328-333

  • Line 276-277: the sentence needs to be better clarified.

Response: The sentence was re-written hoping to improve the quality of the redaction: “Figure 8 A-F, shows how E. coli and P. aeruginosa are forming the bacterial biofilm along the PCL/CUR4 fibers during the 12 h of incubation. As other studies have shown, Gram-negative P. aeruginosa and Gram-positive S. epidermidis bacteria, have displayed uninhibited suspended growth in medium exposed to PCL scaffolds after 6 h at 37°C, moreover both bacterial species multiplied prolifically and populated scaffold surfaces within dense colonies [33].”

Location: Page 14; lines 386-391

  • Figure 8: why is the data of PCL / CUR4 only shown? It is not always certain that the best effects are obtained with the highest concentrations.

Response: Thank you for your comment and we agreed. We added the following explanation in the manuscript before results description as follows:

“This assay was performed as a qualitative test to observe the efficiency of the fibers as bacterial scaffolds for biofilm formation, and thus, its complementary to the above bacterial growth assay. For that reason and taking into account the above results, where no statistical significance difference in the cellular growth was found and no much incidence between concentrations of CUR were definitive in the results. PCL/CUR4 fibers were chosen to show bacterial biofilm formation along the fibers due to its highest quantity of CUR and best appreciation of bacterial cells posed on the fibers. However, the SEM images of PCL, PCL/CUR1, PCL/CUR2 and PCL/CUR3 are presented as supplemental information (Figures S3).

Location: Page 14; lines 378-385

Also, figure S2 was added as supplemental material. Because we belief that the addition of all of the samples in the manuscript will encumber the allowed space in the article page.

Figure S3. SEM images of exposed PCLc and PCL/CUR scaffolds with bacteria after 12 h of incubation. A) Escherichia coli on PCLc fibers (500 x). B) Pseudomona aeruginosa on PCLc fibers (3000 x). C) Staphylococcus aureus on PCLc fibers (3,000 x). D) Escherichia coli on PCL/CUR1 fibers (1000 x). E) Pseudomona aeruginosa on PCL/CUR1 fibers (1600 x). F) Staphylococcus aureus on PCL/CUR1 fibers (3,000 x). G) Escherichia coli on PCL/CUR2 fibers (4000 x). H) Staphylococcus aureus on PCL/CUR2 fibers at (6000 x). I) Staphylococcus aureus on PCL/CUR2 fibers (3,000 x). J) Escherichia coli on PCL/CUR3 fibers (12000 x). K) Staphylococcus aureus on PCL/CUR3 fibers at (6000 x). L) Staphylococcus aureus on PCL/CUR3 fibers (3,000 x).

Location: Supplemental material; figure S3

Discussion section:

  • lane 381: the sentence needs to be better clarified.

Response: thank you for this observation, the following paragraph was re-written as follows:

“Several studies informed about the bioactive properties of CUR which provoked membrane damage in all the tested microorganisms [65]; with concentrations of 25, 50 and 100 µM of CUR I, and after 30 min of exposition, between ~10-20% of the cell viability were affected in most of the tested strains.”

Location: Page 17; lines 502-505

Reviewer 2 Report

The manuscript is devoted to the actual problem being of high importance at biomedical device fabrication and in the food industry. Nevertheless, it can be recommended for the publication (if it at all is possible) only after major revision.

There are view reasons for such a conclusion:

  1. I found no novelty in the manuscript. Opposite, many times I met the sentence like this “It was reported, that ..., which correspond to our results”, or something similar in the meaning. By other words, the author results are similar to the known ones before.
  2. It is absolutely unclear, why the electrospun mat is better for the declared goals compare to cast films fabricated from the same PCL/CUR mixture. No comparison of these two opportunity are presented. The reason of such a question is that only the surface of electrospun mats is used for the bacterial biofilm formation, whereas the internal structure of these mats is not of any importance.
  3. It is absolutely unclear, what type of information (being helpful for biofilm fabrication) can be obtained by FTIR, 1H NMR and DSC analysis methods. Moreover, only a spectra description is presented in the manuscript, whereas no conclusions regarding the state of the material and its promotion on the bacterial biofilm formation.
  4. The properties of 4 different PCL/CUR mixtures are examined in the manuscript. But, no real difference in their properties are observed, and, therefore, I see no reasons for any conclusion.

This list can be continued, but I see in this no need.

The final conclusion is: before discuss the publication possibility, the manuscript should be seriously modified.

Author Response

Dear Reviewer

On behalf of all the authors, we want to thank you and the other reviewers for the time in revising this manuscript, we are grateful for the opportunity to improve its quality. Below you will find the responses and corrections of each comment and suggestions of the 3 reviewers. We appreciated the quality of the reviewer’s comments and we strongly believe that the quality of this manuscript has been improved, hopefully being enough for publication in this important journal. Some of the highlights of this manuscript are as follows:

  • PCL/CUR scaffolds successfully improved the formation of Escherichia coli, Staphylococcus aureus and Pseudomona aeruginosa biofilms.
  • PCL/CUR fibers possess high temperature resilience with degradation temperatures over > 350 oC adequate for bioreactor purposes
  • Is important to control and manipulated biofilm formation, for its study and to better understanding to design of standardized methods to avoid biofilm presence in medical devices, or to use harvested bacteria biofilms for use in biotechnological approaches
  • PCL fibers successfully improved biofilms formation of the three tested bacteria, thanks to the strategy of using a natural bactericide CUR as cellular stressing stimuli.
  • In the comparison study between fibers and films, the results showed how PCL/CUR fibers substantially enhanced the bacterial proliferation compared to the PCL/CUR films.

Note: You can find your comments and responses below attached in PDF with images

Dr. Luis Jesús Villarreal Gómez

__________________________________________________-

Reviewer 2

The manuscript is devoted to the actual problem being of high importance at biomedical device fabrication and in the food industry. Nevertheless, it can be recommended for the publication (if it at all is possible) only after major revision.

There are view reasons for such a conclusion:

  1. I found no novelty in the manuscript. Opposite, many times I met the sentence like this “It was reported, that ..., which correspond to our results”, or something similar in the meaning. By other words, the author results are similar to the known ones before.

Response: Thank you for your comment. We are trying with all of the above expressions to compare/contrast and discuss our findings with others reported in literature. Our purpose is not to say the results of the manuscript is equal to theirs reported, but the obtained results are expected. The novelty and highlights of this work are as follows:

  • PCL/CUR scaffolds successfully increased the formation of Escherichia coli, Staphylococcus aureus and Pseudomona aeruginosa
  • PCL/CUR possess high temperature resilience with degradation temperatures over > 350 oC adequate for bioreactor purposes
  • For future work, these biofilms of specific bacteria can produce metabolites that can be extracted and used in different areas of industry.
  • Is important to control and manipulated biofilm formation, for its study and better understanding to design standardized methods to avoid biofilm presence in medical devices or used harvest bacteria biofilms for its use in biotechnological approaches
  • PCL fibers successfully enhanced biofilm formation of the three tested bacteria, thanks to the strategy to use a natural bactericide CUR as cellular stressing stimuli.
  • In the comparison study between fibers and films. It was demonstrated how PCL/CUR fibers promote substantially the bacterial proliferation compared to the PCL/CUR films.

In the same way the following sentence is added to the manuscript to explain the novelty:

“Moreover, few reports that proposed the use of nanofibers as bacterial biofilms appeared in literature; i.e. electrospun cellulose acetate nanofiber membranes for Lactobacillus plantarum (L. plantarum) present a biofilm formation [32]. In the present study, we will show the capacity of poly (caprolactone)/curcumin (PCL/CUR) to serve as tridimensional scaffolds, that improved the biofilm formation of Staphylococcus aureus, Escherichia coli, Pseudomonas aeruginosa for biotechnological purposes.”

Location: Page 3; lines 75-80

The aim of incorporating a natural bactericidal component like CUR [70] into the inert polymeric nanofibers of PCL, is to stimulate bacterial biofilm by provoking cellular stress in the tested bacteria [6].

Location: Page 18, lines 534-536

“our PCL/CUR tridimensional scaffolds mean to generate the above mechanical stress to bacterial cells; whilst the antimicrobial effect of the CUR, can increased the reactive oxygen species (ROS) and inhibited electron transport [44]”

Location: Page 18; lines 544-546

  1. It is absolutely unclear, why the electrospun mat is better for the declared goals compare to cast films fabricated from the same PCL/CUR mixture. No comparison of these two opportunities are presented. The reason of such a question is that only the surface of electrospun mats is used for the bacterial biofilm formation, whereas the internal structure of these mats is not of any importance.

Response: Thank you for your comment. It has been extensively reported that electrospun nanofibers create a tridimensional architecture that increases the relation surface area-ratio and hence the contact area between bacteria cells and the surfaces of the fibers in the outer and inner mat.

Some ideas are given in the manuscript as follows: “Some desirable characteristics are a tridimensional structure, high surface area, nutrient disposition availability, and controlled degradation of fibers [30]. In a previous work, we reported electrospun nanofibers that can be used for the growth of pharmaceutical drugs producing bacteria, such as actinomycetes [31].”

Location: Page 3; lines 72-75

“our PCL/CUR tridimensional scaffolds mean to generate the above mechanical stress to bacterial cells; whilst the antimicrobial effect of the CUR, can increased the reactive oxygen species (ROS) and inhibited electron transport [40]

Location: Page 18; lines 544-546

Moreover, in order to improve the quality of the assay, we repeated the experiment taking into account the evaluation different proportions of CUR alone and conjugated with PCL in fibers and casting films.

We added the following:

“2.4. Preparation of PCL/CUR cast films

PCL and PCL/CUR casting films were prepared using PCL and PCL/CUR solutions obtained in section 2.2. The membranes were obtained by casting 1.5 mL of each solution in a circular mold (diameter 2.5 cm), followed by the evaporation of solvent at room temperature according to Brianezi, et al., [35].”

  1. Brianezi, S. F. S.; Castro, K. C.; Piazza, R. D.; do Socorro Fernandes Melo, M.; Pereira, R. M.; Marques, R. F. C.; Campos, M. G. N. Preparation and Characterization of Chitosan/MPEG-PCL Blended Membranes for Wound Dressing and Controlled Gentamicin Release. Mater. Res. 2018, 21 (6), e20170951. https://doi.org/10.1590/1980-5373-mr-2017-0951.

Location: Page 4, lines 108-113, reference 35

“Nanofibers mats and cast films samples were cut with a 0.5 cm diameter using a drill and sterilized on both sides with UV-light radiation for 15 min, and placed at the bottom of a 96 plate-wells.”

Location:  Page 5, lines 144-146

 “As a negative control, clean media, clean media with fibers and films were used. As a positive control, 150 μL of medium and 50 μL of each strain were placed without fibers or films.”

Location: Page 5, lines 152-154

“Bacterial growth studies are important to determine whether PCLc, and PCL/CUR fibers better increased bacterial proliferation, compared to PCLc and PCL/CUR films”

Location: Page 11, lines 278, 279

In order to better explain the obtained results; the percentage of proliferation rate was calculated, taking into account the normal growth of each bacteria (bacterial suspension without fibers and films). Where O.D. is the optical density obtained at 620 nm after the exposed time.

                   (1)

Comparing the efficiency in the increment of the bacterial proliferation between PCL/CUR fiber and PCL/CUR films; for Escherichia coli, after 30 min of incubation both fibers and films delay the normal growth, ~7% for fibers and ~30% for films, respectively. Then, after the 12 h, around ~30% of improved growth was observed in the presence of fibers and a huge reduction of growth of about ~ 73% less was elicited from the films. The same behavior was observed after 24 h of incubation.

Hence, for E. coli, PCL/CUR nanofibers clearly increment the proliferation rate and PCL/CUR films alter the normal growth. When comparing between each individual formulation of films and fibers; the most efficient fibrous scaffold was PCL/CUR3, which increased the growth about 37% higher than the normal growth in 12 and 24 h; following with the PCLc fibers which enhanced growth about a 32% at 24 h. The less interesting fibrous scaffold was PCL/CUR1 with just 5% of more growth than the control at 24 h.

No clear difference can be seen between PCL/CUR fibers, with respect to its bioactive effect, when the concentration of CUR was varied. This last result is valuable to assess the appropriate concentration of CUR, depending on the application and expected results. necessity to use certain concentration of CUR into the fibers to obtain the expected results.

 In the case of Pseudomona aeruginosa growth, a different bacterial behavior was observed compare to the latter results. In general, PCL/CUR films didn’t affect bacterial growth at any time interval tested (ANOVA P < 0.05). On the contrary, in contrast PCL/CUR fibers delay the growth by ~52% after the first 12 h; but after 24 h all PCL/CUR fibers increase the proliferation rate ~15%, this can be due to an adaptation period. At 24 h, PCL/CUR3 presented the best results with an increment of (~31%), followed by the PCLc fibers with 22% of enhanced growth with respect to the control. As with the PCL/CUR films, PCL/CUR2, PCL/CUR4 do not significatively affect the normal growth of P. aeruginosa.

Finally, PCL/CUR fibers efficiently promoted the growth of Staphylococcus aureus compared to PCL/CUR films. At 0.5 h, PCL/CUR fibers promoted ~16% the proliferation rate, and PCL/CUR films reduced the growth by ~11%. Then, at 12 h, PCL/CUR fibers enhanced about ~80% the cellular replication whilst PCL/CUR films just enhanced ~5%. Lastly, after 24 h, PCL/CUR fibers promoted the growth by ~36% and PCL/CUR films reduced the normal growth by ~14%. The PCL/CUR fibers with the best results was PCL/CUR1 with ~94% of enhanced growth at 12 h, followed by the PCL/CUR3 fibers, which promoted the replication of S. aureus by ~86% at 12 h.

Hence, it can be demonstrated that PCL/CUR fibers are efficient systems that promoted the growth of all the tested bacterial, particularly these fibrous scaffolds were more effective in promoting growth in Staphylococcus aureus and Escherichia coli.  Bacteria Exposed to PCL/CUR films presented different behaviors depending on the bacterial strain. From all of the PCL/CUR fibers, the system with best results was PCL/CUR3, but no statistically difference between CUR concentration was found (ANOVA P < 0.05). The behavior of exposed bacteria toward the PCL/CUR scaffolds is dependent on the bacterial strain. Nonetheless the system with the best results was PCL/CUR3, although no statistically difference between CUR concentration was found (ANOVA P < 0.05).

In the present study, to ensure the sterilization process (15 min UV irradiation for both sides), all PCL/CUR fibers and films were exposed to clean media, to observe the presence of any growth; but no changes in the O.D. was observed. Moreover, to demonstrated that different CUR concentrations have an effect on the bacteria, as well as confirm the capacity of CUR to cause cellular stress; pure CUR ethanolic extract (dried) at different concentrations (CUR1: 5 µM; CUR2: 7 µM; CUR3: 14 µM and CUR4: 27 µM) and pure CUR leaf were tested in presence of the three bacterial strains. It can be observed that the unprocessed CUR leaf does not significantly affect the normal growth of each bacteria (ANOVA P < 0.05); but, dried CUR ethanolic extracts decreased the growth proportional to concentration, therefore, the higher the concentration of CUR employed, the higher the reduction of proliferation is, in all three bacteria at all times, albeit this decrement is not significant different (ANOVA P < 0.05) between CUR concentrations, a tendency is clearly observed (Figure S2).

Figure 7. Comparative study of bacterial growth of exposed bacteria to PCL/CUR fibers and PCL/CUR films. A) Escherichia coli. B) Pseudomona aeruginosa. C) Staphylococcus aureus.

Location: Pages 11-13; lines 285-366, Figure 7, Figure S2

Figure S2. Relative growth of exposed bacteria to different concentrations of CUR. (a) Escherichia coli. B) Pseudomona aeruginosa. C) Staphylococcus aureus

Location: Supplemental material Figure S2

“Comparing the obtained results between PCL/CUR fibers and films; PCL/CUR fibers significantly increased the growth (ANOVA P < 0.05) of Escherichia coli and Staphylococcus aureus at all times (at all incubation times). For Pseudomona aeruginosa growth occurred only after 24.  PCL/CUR films on the other hand, delay the bacterial replication rate of Escherichia coli and not significantly affect the growth of Pseudomona aeruginosa and Staphylococcus aureus. The enhancing properties of bacterial growth are attributed to the tridimensional structure that fibers offer and the higher surface-area contact that these fibrous scaffolds possess [17]. Cells did not only growth over the surface of the fibrous mat, but the porosity of these mats led to the penetration of bacterial cells into the scaffolds. This phenomenon can be clearly seen in figure 8 and figure S3. These affirmations can also be corroborated by comparing the cell size of each bacteria; Escherichia coli is a typical Gram-negative rod bacterium; with cylinder dimensions of 1.0-2.0 µm long cylinder and radius of about 0.5 µm [66]. Pseudomona aeruginosa have been reported to have a size around ~10 μm [67], and about 0.1 to 0.25 μm in width and roughly 1 to 1.5 μm in length for Staphylococcus aureus [68]. PCL fibers created with standard electrospinning technique parameters reported fibers mats with a pore size between 10–45 mm [69]. These data support the idea of bacterial penetration into the tridimensional structure of the PCL/CUR fibrous scaffolds, where bacterial cells proliferate inside the mat; consequently, after exposure to CUR, bacterial cells stressed and synthesized the extracellular molecules producing biofilms.”

Location: Page 17, 18; lines 514-531

  1. Zheng, H.; Ho, P. Y.; Jiang, M.; Tang, B.; Liu, W.; Li, D.; Yu, X.; Kleckner, N. E.; Amir, A.; Liu, C. Interrogating the Escherichia Coli Cell Cycle by Cell Dimension Perturbations. Proc. Natl. Acad. Sci. U. S. A. 2016, 113 (52), 15000–15005. https://doi.org/10.1073/pnas.1617932114.
  2. Deforet, M.; Van Ditmarsch, D.; Xavier, J. B. Cell-Size Homeostasis and the Incremental Rule in a Bacterial Pathogen. Biophys. J. 2015, 109 (3), 521–528. https://doi.org/10.1016/j.bpj.2015.07.002.
  3. Monteiro, J. M.; Fernandes, P. B.; Vaz, F.; Pereira, A. R.; Tavares, A. C.; Ferreira, M. T.; Pereira, P. M.; Veiga, H.; Kuru, E.; Vannieuwenhze, M. S.; Brun, Y. V.; Filipe, S. R.; Pinho, M. G. Cell Shape Dynamics during the Staphylococcal Cell Cycle. Nat. Commun. 2015, 6 (1), 8055. https://doi.org/10.1038/ncomms9055.
  4. Rnjak-Kovacina, J.; Weiss, A. S. Increasing the Pore Size of Electrospun Scaffolds. Tissue Eng. - Part B Rev. 2011, 17 (5), 365–372. https://doi.org/10.1089/ten.teb.2011.0235.

Location: References 66-69

Finally, this work demonstrated that, no statistical difference was observed in the four different formulations of CUR. This means that, it can be decided whether to use the highest or lowest concentration of CUR for the preparation of CUR loaded polymeric fibers for biotechnological applications, anticipating similar results. Nevertheless, PCL/CUR3 fibers are the most promising system. Moreover, it was demonstrated how the use of the PCL/CUR fibrous scaffolds significantly improved the growth of bacteria, compared to the PCL/CUR films, and thus, effective for biofilm formation.

Location: Page 19, lines 591-597

  1. It is absolutely unclear, what type of information (being helpful for biofilm fabrication) can be obtained by FTIR, 1H NMR and DSC analysis methods. Moreover, only a spectra description is presented in the manuscript, whereas no conclusions regarding the state of the material and its promotion on the bacterial biofilm formation.

 Response: Thank you for your comment, the intention of FTIR and HNMR is to detect any presence of CUR in the PCL samples, but this was not achieved due to the low concentration of CUR in samples. The low concentration of CUR in samples was decided because the objective of this work is to use CUR as stress stimuli and promote biofilm formation. If we use higher concentration of CUR, CUR will be detected by FTIR and 1H NMR, but this will provoke that CUR work as antibacterial agent in the fibrous scaffolds, which is an undesired effect for this work. If is necessary, we can move the FTIR and 1H NMR analysis to the supplementary material. But at least these assays give us information about the integrity of the PCL.

We wrote these paragraphs in the manuscript

“The purpose of the FTIR and 1H NMR studies was to demonstrated the presence of CUR in the fibers, but unfortunately no CUR signals were able to appear due to the low concentration of CUR in the samples.  The small concentration of CUR in the samples was necessary, due to the objective of this work, which is the use of CUR as stress stimuli to promote biofilm formation rather than an antimicrobial agent. If higher concentrations of CUR were used, the signals for the CUR could be detected by FTIR and 1H NMR, but this will provoke that CUR work as antibacterial agent in the fibrous scaffolds, which is an undesired effect for this work.”

Location: Page 19; lines 582-588

“The aim of incorporating a natural bactericidal component like CUR [70] into the inert polymeric nanofibers of PCL, is to stimulate bacterial biofilm by provoking cellular stress in the tested bacteria [6].”

Location: Page 18; lines 534-536

In the case of TGA and DSC, which are thermogravimetric analysis, these assays afforded the thermal stability information necessary for bioreactors purposes. DSC in particular, showed us the melting point of the resultant PCL/CUR fibers in order to see at which temperature these scaffolds start their deformation, and see if they can be used in bioreactor systems.

Some ideas are given in the manuscript as follows:

“TGA was used to evaluate the thermal behavior of the PCLc and PCL/CUR fibers; as mass loss, can be determined with temperature increments [38]”

Location: Page 9; lines 242, 243

“All this data evidences how the morphology of PCL and the concentration of CUR into the PCL fibers affect temperature resilience.”

Location: Page 10; lines 260, 261

“DSC was used to determine the melting point temperature (Tm) and degradation temperature (Td) of PCL/CUR fibers [38]”

Location: Page 10; lines 263, 264

“Tm of our PCL fibers is 63 oC and Tm of our PCL/CUR vary between 62-65 oC, it can be seen clearly how Tm increase when CUR concentration is higher, and these temperatures are slightly higher compared to the above reported Tm of PCL. Fortunately, for bioreactor purposes S. aureus, E. coli and P. aeruginosa grow at 37 oC [57]”

Location: Page 17; lines 482-485

“Meanwhile thermogravimetric analysis (TGA and DSC) showed the great thermal stability of fibers, which will be useful for bioreactor systems”

Location: Page 19; lines 589-590

  1. The properties of 4 different PCL/CUR mixtures are examined in the manuscript. But, no real difference in their properties are observed, and, therefore, I see no reasons for any conclusion.

Response: Thank you for the observation, precisely, we tested 4 different PCL/CUR solutions to know whether CUR concentration affected fiber formation and bacterial proliferation or not. As you just mentioned, no statistical difference was observed in the four different concentrations of CUR. This mean that, we can decide whether to use the highest or lowest concentration of CUR for the preparation of CUR loaded polymeric fibers for biotechnological applications, anticipating similar results.

The following conclusion have been added:

“Finally, this work found that, not statistical difference was observed in the four different concentration of CUR. This mean that, it can be decided whether to use the highest or lower concentration of CUR for the preparation of CUR loaded polymeric fibers for biotechnological applications, with expectation of similar results.”

Location: Page 19; lines 591-597

The real difference was the promotion of proliferation between PCL/CUR fibers compared to the PCL/CUR films

Same answer of comment 2

This list can be continued, but I see in this no need.

The final conclusion is: before discuss the publication possibility, the manuscript should be seriously modified.

Response: Thank you for the opportunity to correct and improve the manuscript, your comments and suggestions, alongside the other reviewers, have been very useful in order to increase the quality of the manuscript. Hopefully, with the added assay and the information clarified, this manuscript can comply with the required quality to be recommended for publication

Reviewer 3 Report

Very interesting paper:

Please clarify the comments below:

  • Line 83" Ethanolic CUR solutions were added at different proportions (2.0, 2.5, 5.0, and 10.0%) to PCL solutions", can you mention at what temperature?
  • I group PCL/Cur4, SEM shows some electrospraying, so you need to mention that.
  • with what you have reported in FTIR, why should I believe that there is any curcumin left there? or maybe it's all degraded? you need to show that your curcumin is not degraded.
  • Do you know if curcumin was in the core of your fibers or not? 

Author Response

Dear Reviewer

On behalf of all the authors, we want to thank you and the other reviewers for the time in revising this manuscript, we are grateful for the opportunity to improve its quality. Below you will find the responses and corrections of each comment and suggestions of the 3 reviewers. We appreciated the quality of the reviewer’s comments and we strongly believe that the quality of this manuscript has been improved, hopefully being enough for publication in this important journal. Some of the highlights of this manuscript are as follows:

  • PCL/CUR scaffolds successfully improved the formation of Escherichia coli, Staphylococcus aureus and Pseudomona aeruginosa biofilms.
  • PCL/CUR fibers possess high temperature resilience with degradation temperatures over > 350 oC adequate for bioreactor purposes
  • Is important to control and manipulated biofilm formation, for its study and to better understanding to design of standardized methods to avoid biofilm presence in medical devices, or to use harvested bacteria biofilms for use in biotechnological approaches
  • PCL fibers successfully improved biofilms formation of the three tested bacteria, thanks to the strategy of using a natural bactericide CUR as cellular stressing stimuli.
  • In the comparison study between fibers and films, the results showed how PCL/CUR fibers substantially enhanced the bacterial proliferation compared to the PCL/CUR films.

Note: You can find your comments and responses below attached in PDF with images

Dr. Luis Jesús Villarreal Gómez

_______________________________________________________

Reviewer 3

Very interesting paper:

Please clarify the comments below:

  • Line 83" Ethanolic CUR solutions were added at different proportions (2.0, 2.5, 5.0, and 10.0%) to PCL solutions", can you mention at what temperature?

Response: Thank you for the observation, the CUR solution was added to the PCL solution at room temperature, with constant stirring for 2 h; according to Bui et al.

We added this information as follows:

“Ethanolic CUR solutions were added at different proportions (2.0, 2.5, 5.0, and 10.0%) to PCL solutions at room temperature (~24oC), with constant stirring at 100 rpm for 2 h; according, to Bui, et al., [34]”

And the following reference was added:

  1. Bui, H. T.; Chung, O. H.; Dela Cruz, J.; Park, J. S. Fabrication and Characterization of Electrospun Curcumin-Loaded Polycaprolactone-Polyethylene Glycol Nanofibers for Enhanced Wound Healing. Macromol. Res. 2014, 22 (12), 1288–1296. https://doi.org/10.1007/s13233-014-2179-6.

Location: Page 3; lines 94-96, reference 34

  • I group PCL/Cur4, SEM shows some electrospraying, so you need to mention that.

Response: Thank you for that observation, we added the following sentence:

“In the case of PCL/CUR4 some electrosprayed artefacts can be seen on the mats, this may be due to the small variations of temperature and % humidity in the ambient”

Location: Page 3; lines 201, 202

  • With what you have reported in FTIR, why should I believe that there is any curcumin left there? or maybe it's all degraded? you need to show that your curcumin is not degraded.

Response: Thank you for your comment, the intention of FTIR and HNMR is to detect any presence of CUR in the PCL samples, but this was not achieved due to the low concentration of CUR in samples. The low concentration of CUR in samples was decided because the objective of this work is to use CUR as stress stimuli and promote biofilm formation. If we use higher concentration of CUR, CUR will be detected by FTIR and 1H NMR, but this will provoke that CUR work as antibacterial agent in the fibrous scaffolds, which is an undesired effect for this work. If is necessary, we can move the FTIR and 1H NMR analysis to the supplementary material. But at least these assays give us information about the integrity of the PCL.

We wrote these paragraphs on the manuscript

“FTIR and 1H NMR studies intend to demonstrated the presence of CUR into the fibers, but unfortunately no CUR signals were able to appear due to the low concentration of CUR in the samples.  The low concentration of CUR in samples was necessary because of the objective of this work, which is the use of CUR as stress stimuli to promote biofilm formation. If higher concentration of CUR is used, CUR can be detected by FTIR and 1H NMR, but this will provoke that CUR work as antibacterial agent in the fibrous scaffolds, which is an undesired effect for this work.”

Location: Page 19; lines 582-588

“The aim of incorporating a natural bactericidal component like CUR [70] into the inert polymeric nanofibers of PCL, is to stimulate bacterial biofilm by provoking cellular stress in the tested bacteria [6].”

Location: Page 18; lines 534-536

In the case of TGA and DSC, which are thermogravimetric analysis, these assays (both as complementary assay between them), give us the thermal stability information necessary for bioreactors purposes. DSC in particular, showed us the melting point of the resultant PCL/CUR fibers in order to see at which temperature these scaffolds start its deformation, and see if they can be used in bioreactor systems.

Some ideas are given in the manuscript as follows:

“TGA was used to evaluate the thermal behavior of the PCLc and PCL/CUR fibers; as mass loss, can be determined with temperature increments [38]”

Location: Page 9; lines 242, 243

“All this data evidences how the morphology of PCL and the concentration of CUR into the PCL fibers affect temperature resilience.”

Location: Page 10; lines 260, 261

“DSC was used to determine the melting point temperature (Tm) and degradation temperature (Td) of PCL/CUR fibers [38]”

Location: Page 10; lines 263, 264

“Tm of our PCL fibers is 63 oC and Tm of our PCL/CUR vary between 62-65 oC, it can be seen clearly how Tm increase when CUR concentration is higher, and these temperatures are slightly higher compared to the above reported Tm of PCL. Fortunately, for bioreactor purposes S. aureus, E. coli and P. aeruginosa grow at 37 oC [57]”

Location: Page 17; lines 482-485

“Meanwhile thermogravimetric analysis (TGA and DSC) showed the great thermal stability of fibers, which will be useful for bioreactor systems”

Location: Page 19; lines 590-590

In the case of demonstrating the degradation of the CUR that was loaded on the fibers; Reversed–Phase High Performance Liquid Chromatography (RP–HPLC) is needed according to Naksuriya et al., 2016. Unfortunately, we have no collaboration yet that possess this equipment. But we will look for collaboration to perform this kind of evaluation in future works after the COVID-19 pandemic is over. It’s a great recommendation.

Unfortunately, in our initial revision of literature we din’t include this evaluation because most of the revised research papers does not perform this degradation test. Some examples are given below:

  1. Bui, H. T.; Chung, O. H.; Dela Cruz, J.; Park, J. S. Fabrication and Characterization of Electrospun Curcumin-Loaded Polycaprolactone-Polyethylene Glycol Nanofibers for Enhanced Wound Healing. Macromol. Res. 2014, 22 (12), 1288–1296. https://doi.org/10.1007/s13233-014-2179-6.
  2. Mutlu, G.; Calamak, S.; Ulubayram, K.; Guven, E. Curcumin-Loaded Electrospun PHBV Nanofibers as Potential Wound-Dressing Material. J. Drug Deliv. Sci. Technol. 2018, 43, 185–193. https://doi.org/https://doi.org/10.1016/j.jddst.2017.09.017.
  3. Rüzgar, G.; Birer, M.; Tort, S.; Acarturk, F. Studies on Improvement of Water-Solubility of Curcumin with Electrospun Nanofibers. ABAD J. Pharm. Sci. 2013, 38 (3), 143–149.
  4. Rezaei, A.; Nasirpour, A. Encapsulation of Curcumin Using Electrospun Almond Gum Nanofibers: Fabrication and Characterization. Int. J. Food Prop. 2018, 21 (1), 1608–1618. https://doi.org/10.1080/10942912.2018.1503300.
  5. Shababdoust, A.; Ehsani, M.; Shokrollahi, P.; Zandi, M. Fabrication of Curcumin-Loaded Electrospun Nanofiberous Polyurethanes with Anti-Bacterial Activity. Prog. Biomater. 2017, 7 (1), 23–33. https://doi.org/10.1007/s40204-017-0079-5.

And more…

In all of the above articles, curcumin don’t lose any bioactivity after its loading in electrospun nanofibers. 

But we are open for increase the quality of the future manuscript to include this evaluation to the set of experiments. Thank you again for your recommendation. As we are concern about the limitation of our study in this regard, the following sentence was added to the manuscript

Another limitation of this study is that the degradation of CUR was not evaluated, a Reversed–Phase High Performance Liquid Chromatography (RP–HPLC) test is needed, according to Naksuriya et al., [53]. Nevertheless, in many studies CUR has been loaded on electrospun nanofibers, none of them reported a loss of bioactivity and/or degradation of the aforementioned compound [10, 11, 13, 14, 34].”

The following reference to support the idea are as follows:

  1. Naksuriya, O.; Vansteenbergen, M. J.; Torano, J. S.; Okonogi, S.; Hennink, W. E. A Kinetic Degradation Study of Curcumin in Its Free Form and Loaded in Polymeric Micelles. AAPS J. 2016, 18 (3), 777–787. https://doi.org/10.1208/s12248-015-9863-0.

Location: Page 16; lines 442-446; reference 53

  • Do you know if curcumin was in the core of your fibers or not? 

Response: Thank you for your question, since we used a single blending electrospinning with a standard set up, it is expected that CUR is mixed homogeneously along the polymeric fibers.

We included the following idea on the manuscript

“As a final note, as is seen in figure 2, it is expected that the CUR distribution on fibers is homogeneously blended along the polymeric fibers, because direct blending electrospinning with a standard set up was used [36, 37].”

Location: Page 7; lines 221-223, references 37, 38

  1. Begum, Hosne Ara and Khan, M. K. R. Study on the Various Types of Needle Based and Needleless Electrospinning System for Nanofiber Production - Google Search. Int. J. Text. Sci. 2017, 6 (4), 110–117. https://doi.org/10.5923/j.textile.20170604.03.
  2. Xue, J.; Wu, T.; Dai, Y.; Xia, Y. Electrospinning and Electrospun Nanofibers: Methods, Materials, and Applications. Chem. Rev. 2019, 119 (8), 5298–5415. https://doi.org/10.1021/acs.chemrev.8b00593.Electrospinning.

Round 2

Reviewer 1 Report

the authors answered my questions and followed my suggestions

Reviewer 3 Report

Thank you for your extensive and reasonable responses.